# Hypoxic pulmonary vasoconstriction as a regulator of alveolar-capillary oxygen flux: A computational model of ventilation-perfusion matching

**Andrew D. Marquis**[1], **Filip Jezek**[1], **David J. Pinsky**[1,2], **Daniel A. Beard**[1¤]*

**1** Department of Molecular and Integrative Physiology, University of Michigan, Ann Arbor, Michigan, United States of America, **2** Department of Internal Medicine, Cardiovascular Medicine, University of Michigan, Ann Arbor, Michigan, United States of America

¤ Current address: Department of Molecular and Integrative Physiology, University of Michigan, Ann Arbor, Michigan, United States of America
* beardda@umich.edu

**Data Availability Statement:** Matlab code for the model can be found at the github repository: https://github.com/beards-lab/VQmodel.

## Abstract

The relationship between regional variabilities in airflow (ventilation) and blood flow (perfusion) is a critical determinant of gas exchange efficiency in the lungs. Hypoxic pulmonary vasoconstriction is understood to be the primary active regulator of ventilation-perfusion matching, where upstream arterioles constrict to direct blood flow away from areas that have low oxygen supply. However, it is not understood how the integrated action of hypoxic pulmonary vasoconstriction affects oxygen transport at the system level. In this study we develop, and make functional predictions with a multi-scale multi-physics model of ventilation-perfusion matching governed by the mechanism of hypoxic pulmonary vasoconstriction. Our model consists of (a) morphometrically realistic 2D pulmonary vascular networks to the level of large arterioles and venules; (b) a tileable lumped-parameter model of vascular fluid and wall mechanics that accounts for the influence of alveolar pressure; (c) oxygen transport accounting for oxygen bound to hemoglobin and dissolved in plasma; and (d) a novel empirical model of hypoxic pulmonary vasoconstriction. Our model simulations predict that under the artificial test condition of a uniform ventilation distribution (1) hypoxic pulmonary vasoconstriction matches perfusion to ventilation; (2) hypoxic pulmonary vasoconstriction homogenizes regional alveolar-capillary oxygen flux; and (3) hypoxic pulmonary vasoconstriction increases whole-lobe oxygen uptake by improving ventilation-perfusion matching.

## Author summary

The relationship between regional ventilation (airflow) and perfusion (blood flow) is a major determinant of gas exchange efficiency. Atelactasis and pulmonary vascular occlusive diseases, such as acute pulmonary embolism, are characterized by ventilation-perfusion mismatching and decreased oxygen in the bloodstream. Despite the physiological

**Funding:** This study was supported by National Institutes of Health grants T32-HL125242, 1F31HL149277-01A1, R01HL150392, and R01-HL139813. The funders had no role in study design, data collection and analysis, decision to publish, or preparation of the manuscript.

**Competing interests:** The authors have declared that no competing interests exist.

and medical importance of ventilation-perfusion matching, there are gaps in our knowledge of the regulatory mechanisms that maintain adequate gas exchange under pathological and normal conditions. Hypoxic pulmonary vasoconstriction is understood to be the primary regulator of ventilation-perfusion matching, where upstream arterioles constrict to direct blood flow away from areas that have low oxygen supply, yet it is not understood how this mechanism affects oxygen transport at the system level. In this study we present a computational model of the ventilation-perfusion matching and hypoxic pulmonary vasoconstriction to better understand how physiological regulation at the regional level scales to affect oxygen transport at the system level. Our model simulations predict that this regulatory mechanism improves the spatial overlap of airflow and blood flow, which serves to increase the uptake of oxygen into the bloodstream. This improved understanding of ventilation-perfusion matching may offer insights into the etiology of, and therapeutic interventions for diseases characterized by ventilation-perfusion mismatching.

## Introduction

The pulmonary circulation exhibits several unique features when compared to systemic circulation. In a healthy individual, mean systemic arterial blood pressure tends to be between 90-110 mmHg, whereas mean pulmonary arterial pressure tends to be between 15-20 mmHg. Since the lungs are perfused with the entire cardiac output, they are perfused with the largest flow at the lowest perfusion pressure of any organ. At the regional level, flow in the pulmonary circulation is regulated via unique mechanisms. In response to hypoxia, systemic arteries tend to dilate. At the whole-body level this response is mediated by neural mechanisms such as the central and peripheral chemoreflex [1]. At the local level, metabolism-mediated vasodilating factors are released in response to increasing oxygen demand. One potential metabolic factor is extracellular ATP, which can act as a vasodilator through the activation of P2Y2 receptors [2, 3]. Pulmonary arteries however, constrict in response to low oxygen in the alveolar space [4, 5]. This phenomenon is referred to as hypoxic pulmonary vasoconstriction (HPV), and is believed to operate on a regional level by directing blood flow away from airway regions that have low oxygen supply [5].

Efficient pulmonary gas exchange requires effective matching of airflow and blood flow—referred to as ventilation-perfusion (V/Q) matching. In large mammals, such as humans, V/Q ratios are significantly influenced by gravity, resulting in increased blood flow through the lowest part of the organ [6]. The weight of the organ and extravascular tissue compresses the parenchyma [7]. By extension, a subject's posture (ex: supine, prone, upright) has appreciable consequences on regional perfusion and V/Q ratios [8]. Moving a patient into a prone position has been shown to improve V/Q matching and oxygenation [9]. However, very little is known about how V/Q matching is *actively* regulated through physiological regulatory mechanisms. While the ability of central and peripheral chemoreflexes to alter the rate and depth of the ventilation for the whole organ is well appreciated, regulation at the regional-level is not understood [1]. Similarly, bronchoconstriction and bronchodilation of the airway have been well studied and discussed in the context of asthma [10, 11]. There are a few experimental studies which suggest that hypocapnic bronchoconstriction could be an important regulatory mechanism for V/Q matching, but the evidence has not definitely proven the role of this mechanism [12, 13]. To our knowledge there is no evidence to suggest that the small airways sense airway oxygen and physiologically regulate the distribution of air flow [14, 15]. HPV is understood to be the major homeostatic feedback mechanism which physiologically maintains adequate V/Q

matching. Chronic exposure to high altitude and inappropriate activation of HPV have also been implicated in the development of pulmonary arterial hypertension [16–19].

The molecular basis of HPV remains elusive. A common perspective is that the oxygen sensor, transducer, and vasoconstrictive effector must all be contained within pulmonary arterial smooth muscle cells [17]. From this perspective, several pathways including modulation of endothelin signaling, and regulation of prostacyclin synthesis have been proposed as the molecular integrators of HPV. However, functional studies investigating these specific pathways have been unable to demonstrate their role in mediating the phenomenon [20]. Kuebler and coworkers have proposed a signal transduction pathway for HPV wherein an oxygen-sensitive sphingomyelinase and the cystic fibrosis transmembrane regulator (CFTR) induce the translocation of TRPC6 calcium channels to the membrane of vascular smooth muscle cells in the smallest arterioles that perfuse the alveoli [21]. In this mechanism, membrane depolarization initiates at the alveolocapillary level, and a conducted wave of depolarization then propagates along the endothelium in the opposite direction of blood flow through connexin-40 gap junctions [22]. Dunham et al. have also recently proposed a pathway for HPV in which hydrogen peroxide generated by nicotinamide adenine dinucleotide dehydrogenase iron-sulfur protein 2 (Ndufs2—a core subunit of mitochondrial complex 1) regulates the activity of Kv1.5 ion channels on pulmonary arterial smooth muscle cells [23].

A popular perspective in the V/Q matching literature is that HPV is not important under normoxic conditions as major changes in pulmonary arterial pressure or total pulmonary vascular resistance are generally only observed when alveolar oxygen falls below 80 mmHg. This view is supported by experiments in Arai et al [24]. The authors hypothesized that some of the heterogeneity in pulmonary blood flow present in the normal human lung in normoxia is due to HPV. To test this, they exposed human subjects to mild hyperoxia and measured blood flow heterogeneity via arterial spin labeling. Their results show that under mild hyperoxia that there was little change in the distribution of blood flow compared to normoxic conditions, and conclude that HPV does not affect pulmonary perfusion heterogeneity in normoxia. We are skeptical of hyperoxia exposure being an effective perturbation to uniformly inhibit HPV, and the conclusions of this study. Under normoxic conditions there is heterogeneity in the distribution of airflow and oxygen in the airway. This means that there are likely some regions of lung where HPV could be activated. Under hyperoxic conditions the average oxygen content in the airway certainly increases, however the heterogeneity of the airway oxygen distribution is likely unchanged—as demonstrated by experiments where airflow distribution is measured when a pig is ventilated on different fractions of inspired oxygen [15]. Given that HPV operates to alter the blood flow distribution to mirror the airflow distribution, and that hyperoxia exposure does not alter the distribution of airflow, we assert that the role of HPV in affecting blood flow distribution in normoxia remains an open question. Effective inhibition of HPV by transgenic or pharmacological manipulation could potentially decouple the oxygen sensor from the vasoconstrictive effector. Our skepticism of global hyperoxia not being an effective way to inhibit HPV could be reconciled by measuring the distribution of alveolar oxygen tension. However, routine clinical measurements of alveolar oxygen are an indirect extrapolation using an empirical formula based on blood gas measurements and only represent the average alveolar oxygen tension [25]. It is likely that HPV function would be most apparent during hypoxia exposure, conditions where oxygen consumption is increased (cardiovascular exercise), or when there are significant structural occlusions of the vasculature (ex: pulmonary embolism) or airway (ex: inhaled foreign body).

Given the complexity of V/Q matching, there have been many efforts to computationally-model the physiology of pulmonary gas exchange. Compartmental formulations have been useful to scrutinize system-level regulation by neuro-humoral mechanisms, as well as effects of

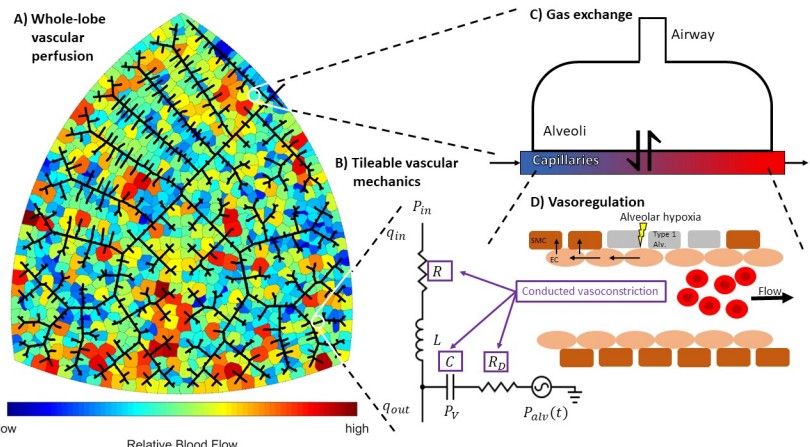

**Fig 1. Schematic of our multi-scale multi-physics model of ventilation-perfusion matching.** Block (A) illustrates the whole-lobe vascular network model. Black lines represent blood vessels, and colored regions represent discrete zones of perfusion. The network geometry is agnostically generated by a space-filling algorithm inspired by Wang et al. [33]. Block (B) shows how the mechanics of each vessel segment is represented as an equivalent circuit. Intravascular pressure ($P_v$) and flow into the vessel ($q_{in}$) are the state variables; inlet pressure ($P_{in}$) and flow out of the vessel ($q_{out}$) are the initial conditions at boundaries for a given vessel segment; outlet pressure ($P_{out}$) is an algebraic constraint; time-varying alveolar pressure ($P_{alv}$) is a dynamic pressure source; and hydraulic resistance ($R$), inertance ($L$), compliance ($C$), and vessel wall resistance ($R_D$) are anatomical parameters calculated from the geometry of the vessel segment (length and radius) and can be modulated by vasoregulation (purple boxes). Block (C) depicts a representative gas exchange unit. Gases flowing through the capillary tube are exchanged with an alveolar compartment. Block (D) portrays the oxygen-sensitive vasoregulatory mechanism hypoxic pulmonary vasoconstriction (HPV). Hypoxia in the alveolar space induces conducted vasoconstriction. Conducted vascular responses are spatially propagated in a upstream through the endothelium—these responses modulate the values of the anatomical parameters ($R$, $C$, and $R_D$) in the arterial network.

binding and buffering of oxygen and carbon dioxide [26, 27]. There are also structural models which have been constructed to investigate the impact of airway and vascular geometry on regional flow distribution [28–30]. These types of models have been critical in informing our current understanding of how gravity and posture impact that distribution of blood flow. These structural models offer a direct way to translate patient-specific geometries to functional predictions. Burrowes et al. model how pulmonary emboli will redistribute blood flow, and the impact of oxygenation of red blood cells [31]. They predict that occluding 80% of the vascular network will increase the mean red blood cell transit time above the time needed to fully saturate the cell's hemoglobin. More recent work from Burrowes et al. use an empirical model HPV on top of their structural model to predict the role of this mechanism in acute pulmonary embolism. [32].

While there are many factors that influence V/Q matching, the goal of this study is to understand how the integrated action of HPV affects V/Q matching and regional alveolar-capillary oxygen flux throughout a lobe of the lungs. We present a multi-scale multi-physics computational model of V/Q matching in rat lungs where the critical components are: (a) an algorithm to generate morphometrically realistic pulmonary vascular networks; (b) a tileable lumped-parameter model of vascular fluid and wall mechanics; (c) oxygen transport accounting for oxygen bound to hemoglobin and dissolved in plasma; and (d) a novel empirical model of HPV based on the conducted vascular response mechanism proposed by Wang et al. [22]. A schematic of our model is shown in Fig 1. The model is calibrated to data available in published literature. Our model simulations predict that under the artificial test condition of a uniform ventilation distribution (1) HPV matches perfusion to ventilation; (2) HPV homogenizes

regional alveolar-capillary oxygen flux; and (3) HPV increases whole-organ oxygen uptake by improving V/Q matching.

## Methods

### Model

Our computational work has been implemented in MATLAB and the corresponding codes have been made available in the GitHub repository https://github.com/drewzeewoozee/VQmodel. Table 1 contains a list of all model variables, fixed parameters, and adjustable parameters.

The following subsections explain how we:

- Generate morphometrically realistic vascular networks

- Derive equations that govern vascular fluid mechanics

- Derive equations that govern oxygen transport

- Derive equations of an empirical model of HPV

The major novel features are the empirical model of HPV, and the integration of these sub-models into a multi-scale multi-physics framework that enables us to understand how the regulation of blood flow distribution impacts oxygen transport.

**Distributive branching networks.** To generate the topology of vascular networks we adapt the angiogenesis-inspired distributive branching algorithm of Wang et al. [33]. Wang et al.'s original algorithm requires a domain of mesh points for the network to span and uses a purely dichotomous branching mechanism to "fill" the space bounded by the mesh points. Our algorithm similarly requires a user supplied domain of points. We generated various shaped 2-dimensional domains of mesh points by using a finite-element meshing algorithm (PolyMesher) [34]. Each iteration of our algorithm adds a new node to the network. Network nodes can be one of four types: 0—the network "root"—there is only one of these in a given network; 1—"normal" nodes which have a single parent and a single daughter; 2—"bifurcation" nodes which have a single parent and two daughters; and 3—"terminal" nodes which have a single parent and no daughters. Our algorithm in pseudo-code is illustrated in Fig 2. This algorithm requires user inputs of $\Delta g$—the grow size distance for each node, $K$—the number of nodes to generate, $x$ and $y$—the x and y coordinates of the mesh points, and $r$—a $2 \times 1$ vector specifying the location of the root node.

The output of our algorithm is an undirected graph which describes the topological connectedness of a vascular network. To interpret each branch of the network as a vessel we assign them a length and radius. This allows us to validate the morphology of our networks by comparing them to experimental data. The data set we compare our networks to reports the length and radius of rat pulmonary arterial networks using the Diameter-defined Strahler ordering system [35]. To make such comparisons, we first determine the vessel segments— network branches in between bifurcation nodes. We assign a radius to each vessel segment based on Poiseuille's law:

$$\Delta P = \frac{8\mu(\text{Hct}, r)lQ}{\pi r^4} \tag{1}$$

where $\Delta P$ is the pressure drop across the segment, $\mu$ is the viscosity of blood, $l$ is the length of the segment, $Q$ is the volumetric flow rate, and $r$ is the radius of the segment. We use Pries et al.'s nonlinear viscosity law [36] for $\mu(\text{Hct}, r)$ to account for the rheological properties of blood flow in the microcirculation (Fahraeus and Fahraeus-Lindqvist effects). We set the flow

**Table 1. Model variables, fixed parameters, and adjustable parameters.** Values are omitted if they are time varying or defined for each vessel segment. For the adjustable parameters estimated by the BELUGA Genetic Algorithm, values are reported for each network (A,B,C).

**Variables**

| Name | Units | Interpretation | Estimator | Reference | Value |
|---|---|---|---|---|---|
| $q_{in}$ | ml/s | flow rate into a vessel segment | - | - | - |
| $q_{out}$ | ml/s | flow rate out of a vessel segment | - | - | - |
| $P_v$ | mmHg | wall pressure of a vessel segment | - | - | - |
| $P_{in}$ | mmHg | inlet pressure of a vessel segment | - | - | - |
| $P_{out}$ | mmHg | outlet pressure of a vessel segment | - | - | - |
| $V$ | ml | volume of blood in a vessel segment | - | - | - |
| $C_{O2}$ | mol/L | concentration of $O_2$ in a vessel segment | - | - | - |
| $P_{O2}$ | mmHg | partial pressure of $O_2$ in a vessel segment | - | - | - |
| $V_{alv}$ | ml | volume of air in an alveoli | - | - | - |
| $M_{alv}$ | mol | mass of $O_2$ in an alveoli | - | - | - |
| $P_{alv,O2}$ | mmHg | partial pressure of $O_2$ in an alveoli | - | - | - |
| $P_{airway,i}$ | mmHg | partial pressure of $O_2$ in the $i$'th airway | - | - | - |
| $T$ | unitless | vasoactive tone in a vessel segment | - | - | - |

**Fixed Parameters**

| Name | Units | Interpretation | Estimator | Reference | Value |
|---|---|---|---|---|---|
| $l$ | cm | vessel segment length | from network | [38] | - |
| $r$ | cm | vessel segment radius | Eq (1) | [38] | - |
| $R_0$ | mmHg s/ml | nominal vessel segment resistance | $R_0 = \frac{8\mu(r,\text{Hct})l}{\pi r^4}$ | [38] | - |
| $E$ | mmHg | elastic modulus of a blood vessel | - | [43] | 10,000 |
| $h$ | mm | vessel wall thickness | $h = r(ae^{br} + ce^{dr})$ | [43] | - |
| $a$ | mm | empirical constant for $h$ estimator | - | [43] | 0.2802 |
| $b$ | mm | empirical constant for $h$ estimator | - | [43] | -0.5053 |
| $c$ | mm | empirical constant for $h$ estimator | - | [43] | 0.1324 |
| $d$ | mm | empirical constant for $h$ estimator | - | [43] | -0.01114 |
| $C_0$ | ml/mmHg | Nominal vessel segment compliance | $C_0 = \frac{\pi r^3 l}{Eh}$ | [38] | - |
| $\nu$ | s | vessel relaxation speed | - | [38] | 0.01 |
| $R_D$ | mmHg s/ml | vessel wall resistance | $R_D = \frac{\nu}{C}$ | [38] | - |
| $\rho$ | kg/m$^3$ | density of blood | - | [38] | 1060 |
| $L$ | mmHg s$^2$/ml | vessel segment intertance | $L = \frac{\rho l}{\pi r^2}$ | [38] | - |
| $\alpha$ | mol/L/mmHg | $O_2$ solubility in plasma | - | [44] | $1.3 \times 10^{-6}$ |
| $\beta$ | mmHg/L/mol | $O_2$ solubility in air | - | [44] | $5.95 \times 10^{-5}$ |
| $C_{Hb}$ | mol/L | Oxyhemoglobin binding site concentration | - | [45] | 0.0213 |
| Hct | unitless | hematocrit—fraction of blood made out of RBCs | - | - | 0.4 |
| $n$ | unitless | Hill exponent for Hb saturation | - | [46] | 2.6 |
| $P_{50}$ | mmHg | partial pressure of $O_2$ where Hb is half saturated | - | [46] | 36 |
| $C_{O2,input}$ | mol/L | $O_2$ entering pulmonary circulation | - | - | $7.84 \times 10^{-3}$ |
| $R_R$ | 1/s | respiratory frequency | - | - | 1.17 |
| $Q_P^*$ | ml/s | perfusion magnitude | $Q_P^* = Q_{P,\text{Cheng}}\frac{CO_{\text{Rat,ref}}}{CO_{\text{Human,ref}}}$ | [47] | 0.66 |
| $Q_V^*$ | ml/s | ventilation magnitude | $Q_V^* = Q_{V,\text{Cheng}}\frac{CO_{\text{Rat,ref}}}{CO_{\text{Human,ref}}}$ | [47] | 2.97 |
| $P_{alv}^*$ | mmHg | alveolar pressure amplitude | - | - | 0.75 |
| $P_{HPV}$ | mmHg | HPV $O_2$ sensitivity | $P_{HPV} = \frac{80\ \text{mmHg}}{\ln(2)}$ | Eq (38) | 115 |
| $\lambda$ | mm | HPV signal spatial decay constant | - | [42] | 100 |
| $\tau$ | s | HPV time constant | arbitrarily fixed | - | 30 |

(*Continued*)

**Table 1.** (Continued)

| Adjustable Parameters | | | | | |
|---|---|---|---|---|---|
| Name | Units | Interpretation | Estimator | Reference | Value |
| $D_{O2}$ | ml/s | apparent alveolar-capillary $O_2$ diffusivity | BELUGA GA | [48] | (3.36, 14.60, 0.95) |
| $T_{Max}$ | mm | HPV maximum vessel tone | BELUGA GA | [48] | (1.49, 1.48, 1.49) |
| $T_{min}$ | mm | HPV minimum vessel tone | BELUGA GA | [48] | (0.49, 0.48, 0.31) |

rate in each segment by first determining the area of each perfusion zone. A perfusion zone is defined as the area spanned by mesh points closest to a terminal branch in the network—conceptually similar the Voronoi diagram [37]. We assign a flow through a terminal segment proportional to the area it perfuses. Using the topology of the network, we back-calculate the flow in each segment of the network. To estimate the pressure drop for each segment, we assume that $\Delta P = ql/k$, where $q$ is the flow through the vessel segment, and $k$ is a constant hydraulic conductivity per unit length ($\mu m^2$/mmHg/s). Eq (1) is numerically solved to determine $r$, the radius of each vessel segment.

Once we have assigned a radius to each vessel segment, we categorize their orders by the diameter-defined Strahler system as detailed in Jiang et al. [35]. We then compare the length and radius of each Strahler order in our vascular networks to Jiang et. al.'s morphometric data. We manually calibrate our network anatomy to the morphometric data by adjusting the size and shape of the finite-element meshes generated by Polymesher, and adjusting the value of $k$ to achieve reasonable fits to the morphometric data in Fig 3.

**Vascular mechanics.** We represent fluid mechanics through the vascular network using a tileable lumped-parameter model following the bond graph approach of Safaei et al. [38]. Fig 1B shows how the mechanics of each vessel segment may be represented as an equivalent RC electric circuit. The wall of each vessel segment is equivalent to Voigt body model. For each

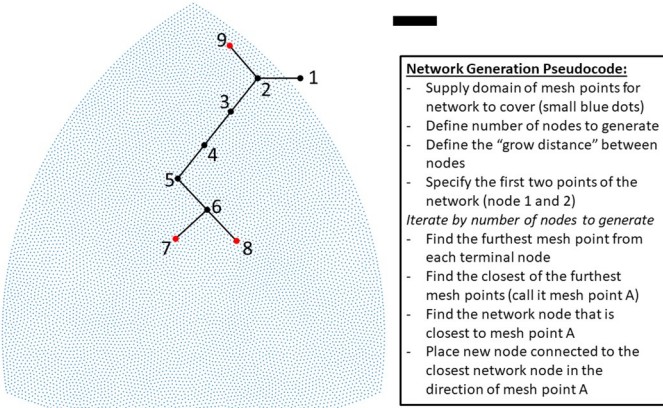

**Fig 2. Example of angiogenesis-inspired network generation algorithm.** Blue points define a mesh that the algorithm seeks to cover with a network. The black and red dots are the nodes generated by each iteration of the algorithm; red nodes denote network terminals; black lines are connections between nodes. The black scale bar is 1000 $\mu$m. The first two nodes (1 and 2) are manually specified by the user. The algorithm determines the mesh points furthest from the network. For nodes 3-5, the southwest corner is the furthest mesh point. For node 6, the southeast corner is furthest and then for node 7 the southwest corner is furthest. For node 8, the southeast corner is the furthest point and is connected to closest node of the network—node 6. The northern most mesh point is furthest from both node 7 and 8. Node 9 is connected to node 2 since it is the closest node in the existing network. To implement the user needs to supply a domain of points to cover, the "grow distance" between nodes, and the number of nodes to generate.

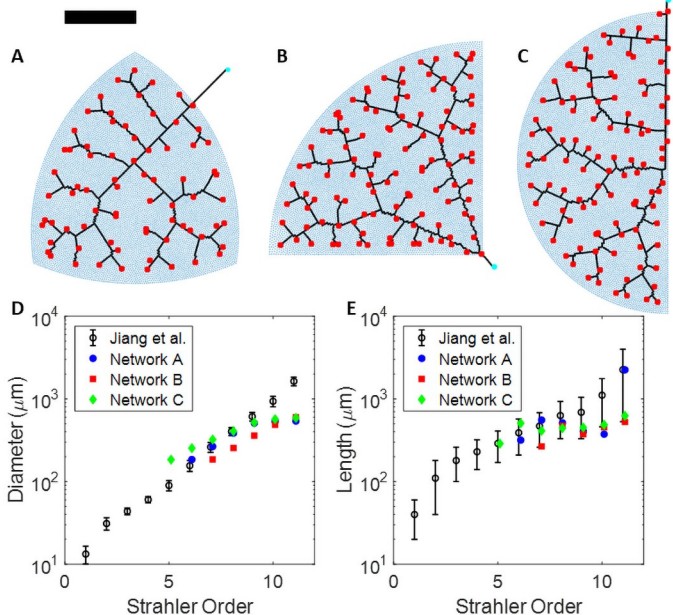

**Fig 3. Morphometrically realistic pulmonary vascular networks.** (A,B,C) Synthetic population of rat lungs. Blue dots are the finite-element mesh points made by Polymesher [34], the cyan circles denote the root node for each network, the black lines represent the network topology, and the red squares denote the network terminals. The black scale bar is 3000 $\mu$m. All networks were made with 1,000 nodes and $\Delta g$ set to 50 $\mu$m. Note that the radius of each vessel is not plotted to size. The total area spanned by each finite element mesh is the same. (D, E) Comparison of our network geometries to measured diameters and lengths as a function of diameter-defined Strahler orders from silicone-elastometer casts of rat pulmonary arterial trees in Jiang et al. [35].

segment, the change in vessel wall pressure is given by

$$C\frac{dP_v}{dt} = q_{in} - q_{out},\tag{2}$$

where $P_v$ is the vessel wall pressure, $C$ is the vessel compliance, and $q_{in}$ and $q_{out}$ are the flow in and out of the vessel respectively. The lumenal outlet pressure of each segment is determined by

$$P_{out} - P_{alv} = P_v + R_D(q_{in} - q_{out})\tag{3}$$

where $P_{alv}$ is the external alveolar pressure, $P_{out}$ is the lumenal vessel outlet pressure, and $R_D$ is the vessel-wall resistance. Inlet flow $q_{in}$ is governed by

$$L\frac{dq_{in}}{dt} + Rq_{in} = P_{in} - P_{out},\tag{4}$$

where $L$ is the blood inertance, $R$ is the hydraulic resistance, and $P_{in}$ is the inlet pressure. Eqs (2)–(4) constitute a system of two ordinary differential equations where $P_v$ and $q_{in}$ are the state variables and $P_{in}$ and $q_{out}$ are the conditions at the boundary of a given vessel segment. We assume hydraulic resistance $R$ is a non-linear quantity that depends on the volume of the vessel segment and is represented by

$$R = R_0\left(\frac{V_0}{V}\right)^2,\tag{5}$$

where $R_0$ is the resistance, $V_0$ is the fluid volume at an idealized equilibrium, and $V$ is the instantaneous volume of the vessel segment, given by

$$V = CP_v,\tag{6}$$

which holds for all vessel segments.

For an arterial bifurcation, flow from a parent segment is partitioned into two daughter segments. $P_{in}$ for the daughter segments is specified as the $P_{out}$ of the parent segment. $q_{out}$ of the parent segment is specified by conservation of flow with

$$q_{out,p} = q_{in,d1} + q_{in,d2},\tag{7}$$

where $p$, $d1$, and $d2$ denote parent and daughter segments respectively. Terminal arterioles connect to capillary segments whose mechanics are represented in the same manner. By conservation of flow and pressure balance we have $q_{out,art} = q_{in,cap}$, and $P_{in,cap} = P_{out,art}$, where $art$ and $cap$ denote arterial and capillary segments. Capillary segments drain into a venous network that is topologically equivalent to the arterial network. Similarly, we have $q_{out,cap} = q_{in,ven}$, and $P_{in,ven} = P_{out,cap}$, where $ven$ denotes the venous segments. For a venous bifurcation, flow through two "partner" segments merges into a confluent segment. Conservation of flow and pressure balance requires

$$q_{in,c} = q_{out,a} + q_{out,b}, \text{ and}\tag{8}$$

$$P_{v,a} + R_D(q_{in,a} - q_{out,a}) = P_{v,b} + R_D(q_{in,b} - q_{out,b})\tag{9}$$

where $a$ and $b$ denote the "partner" segments and $c$ denotes the confluent segment. Since $q_{out}$ is an initial condition, we need to solve for $q_{out,a}$ and $q_{out,b}$. Eqs (8) and (9) can be solved to yield

$$q_{out,a} = \frac{P_{v,a} - P_{v,b} + R_{D,b}q_{in,c} + R_{D,a}q_{in,a} - R_{D,b}q_{in,b}}{R_{D,a} + R_{D,b}},\tag{10}$$

$$q_{out,b} = -\left(\frac{P_{v,a} - P_{v,b} + R_{D,a}q_{in,c} + R_{D,b}q_{in,b} - R_{D,b}q_{in,a}}{R_{D,a} + R_{D,b}}\right).\tag{11}$$

**Airway and vascular oxygen transport.** We treat a single capillary vessel segment interacting with an alveolar compartment as the gas exchange unit. Although pulmonary capillary blood flow resembles a sheet of fluid moving across the alveoli [39, 40], our model does not explicitly represent this spatial feature of the gas exchange unit. Instead we use a lumped compartmental approach and represent the diffusion capacity as an apparent diffusion parameter ($D_{O2}$) that we estimate in the Model Parameterization section.

For the airway, we use a simplified compartmental model to represent oxygen transport which is shown in S1 Fig. We assume fixed volumes for the airways, and variable volumes for the alveolar compartments. Equations in this section are derived based on compartmental mass-balance. In the airway, oxygen content is represented by partial pressure (mmHg). For oxygen dissolved in the blood there is a non-linear relationship between blood oxygen partial pressure and blood oxygen concentration due to oxygen being bound to hemoglobin.

Airway1 represents the proximal airway and interacts with atmospheric air, airway2 connects to airway1, and all the alveolar compartments. Airway2 functions as a mixing compartment and is representative of the anatomical dead-space. The model is driven by two empirical

functions:

$$Q_{air}(t) = Q_V^* \, \sin(2\pi R_R t), \text{ and,} \tag{12}$$

$$P_{alv}(t) = P_{alv}^* (2\pi R_R t) \tag{13}$$

where $Q_V^*$ is the magnitude of air flow (ml/s), $P_{alv}^*$ is the amplitude of the alveolar pressure, and $R_R$ is the number of breaths per second. The oxygen mass balance in airway1 is governed by

$$\frac{dP_{airway1,O2}}{dt} = \begin{cases} \dfrac{Q_{air}}{V_{airway1}}[P_{air,O2} - P_{airway1,O2}] & Q_{air} \geq 0 \\[3mm] -\dfrac{Q_{air}}{V_{airway1}}[P_{airway2,O2} - P_{airway1,O2}] & Q_{air} < 0 \end{cases} \tag{14}$$

where $P_{airway1,O2}$ is the partial pressure of oxygen in airway1, $V_{airway1}$ is the volume of air in airway1, and $P_{air}$ is the partial pressure of oxygen in the atmosphere. Oxygen mass balance in airway2 is governed by

$$\frac{dP_{airway2,O2}}{dt} = \begin{cases} \dfrac{Q_{air}}{V_{airway2}}[P_{airway1,O2} - P_{airway2,O2}] & Q_{air} \geq 0 \\[3mm] -\dfrac{\sum_{i=1}^{M} q_i[P_{alv,O2,i} - P_{airway2,O2,i}]}{V_{airway2}} & Q_{air} < 0 \end{cases} \tag{15}$$

where $P_{airway2,O2}$ is the partial pressure of oxygen in airway2, $M$ is the number of perfusion zones, $q_i$ is the airflow to the $i$'th alveolar compartment (mL/s), and $P_{alv,O2,i}$ is the partial pressure of oxygen in the $i$'th alveolar compartment such that $Q_{air}(t) = \sum_{i=1}^{M} q_i(t)$. For the $i$'th alveolar compartment, change in volume is given by

$$\frac{dV_{alv,i}}{dt} = q_i, \tag{16}$$

or

$$V_{alv,i}(t) = V_{0,i} + \frac{q_i}{2\pi R_R}(1 - \cos(2\pi R_R t)), \tag{17}$$

where $V_{0,i}$ is the initial volume of the $i$'th alveolar compartment. The mass (in moles) of oxygen in the $i$'th alveolar compartment is given by

$$\frac{dM_{alv,i}}{dt} = \begin{cases} \dfrac{q_i}{\beta}P_{airway2,O2} - \alpha D_{O2}[P_{alv,O2,i} - P_{O2,i}] & Q_{air} \geq 0 \\[3mm] \dfrac{q_i}{\beta}P_{alv,O2,i} - \alpha D_{O2}[P_{alv,O2,i} - P_{O2,i}] & Q_{air} < 0 \end{cases} \tag{18}$$

where $\alpha$ is the Henry solubility coefficient for oxygen in a fluid (mol/L/mmHg), $\beta$ is the solubility of oxygen in gas (mmHg L/mol), $D_{O2}$ is the apparent diffusion capacity of oxygen across the alveolar-capillary boundary, $P_{alv,O2,i}$ is the partial pressure of oxygen in the $i$'th alveolar compartment, and $P_{O2,i}$ is the partial pressure of oxygen in the $i$'th capillary vessel segment. The oxygen partial pressure in the alveoli is computed by

$$P_{alv,O2,i}(t) = \beta \frac{M_{alv,i}(t)}{V_{alv,i}(t)}. \tag{19}$$

Eq (18) is cast in units of total mass of oxygen per unit time. Eq (19) is used to calculate the corresponding oxygen partial pressure in the alveolar compartment.

To represent oxygen within the blood, we account for oxygen both freely dissolved in plasma and that bound to hemoglobin, so that

$$C_{O2} = \alpha P_{O2} + \text{Hct} C_{Hb} \left( \frac{P_{O2}^n}{P_{O2}^n + P_{50}^n} \right) \tag{20}$$

where $C_{O2}$ is the concentration of oxygen (total moles per L), $P_{O2}$ is the partial pressure of oxygen (mmHg), $C_{Hb}$ is the average concentration of hemoglobin binding sites in a red blood cell (mol/L), $P_{50}$ is the oxygen partial pressure in which half the hemoglobin binding sites are bound to oxygen, and $n$ is the Hill-coefficient. For the $i$'th capillary compartment, change in oxygen concentration is given by

$$\frac{dC_{O2,i}}{dt} = \frac{q_{in,i}(C_{O2,input} - C_{O2,i}) + \alpha D_{O2}(P_{alv,i} - P_{O2,i})}{V_{cap,i}}, \tag{21}$$

where $q_{in,i}$ is the inlet flow rate for the $i$'th capillary segment, and $V_{cap,i}$ is the volume of the $i$'th capillary segment. We assume that there is no partitioning of hematocrit or uptake of oxygen in the arterial network. For the $j$'th venous segment, change in oxygen concentration is given by

$$\frac{dC_{O2,j}}{dt} = \frac{q_{out,a}(C_{O2,a} - C_{O2,j}) + q_{out,b}(C_{O2,b} - C_{O2,j})}{V_{ven,j}}, \tag{22}$$

where $a$ and $b$ represent the upstream venous segments that become confluent with the $j$'th venous segment, and $V_{ven,j}$ is the instantaneous volume of the $j$'th venous segment.

**Empirical hypoxic pulmonary vasoconstriction.**   Our empirical representation of hypoxic pulmonary vasoconstriction (HPV) is inspired by Wang et al. [22], in which they demonstrate that connexin-40 gap junctions facilitate the transduction of a wave of depolarization from the alveolar-capillary boundary upstream to the arterioles and arteries. For the $i$'th perfusion zone, the HPV signal generated by the oxygen content in the alveoli is represented by

$$S_i = \exp \left[ -\frac{P_{alv,O2,i}}{P_{HPV}} \right] \tag{23}$$

where $S_i$ is a unitless signal, and $P_{HPV}$ is a parameter representing the sensitivity of HPV to oxygen partial pressure in the alveolar space. An implicit assumption built into Eq (23) is that HPV senses and generates a signal across all possible values of $P_{alv,O2,i}$. The exponential form of Eq (23) was chosen so that $S_i$ was strictly positive and decreasing as a function of alveolar oxygen. In other words, the less oxygen in the alveolar space, the greater the HPV signal, $S_i$. The target tone of the $j$'th arterial segment is given by

$$T_{j,\infty} = \left( \sum_{k=1}^{N_k} S_k \, \exp \left[ -\frac{l_k}{\lambda} \right] \right) (T_{Max} - T_{min}) + T_{min}, \tag{24}$$

where $N_k$ is the number of perfusion zones that the $j$'th arterial segment feeds into, $k$ is the index of those perfusion zones, $l_k$ is the path length from the perfusion zone to the mid-line of the $j$'th segment, $\lambda$ is a length constant governing the spatial decay of the transduced signal, and $T_{Max}$ and $T_{min}$ are unitless scaling parameters. The summation in Eq (24) represents the cumulative HPV signal in the $j$-th vessel segment influenced by the $k$-many downstream perfusion zones. The strengths of the HPV signals, $S_k$, are weighted by an exponential spatial decay

of signal strength along the length of the signal. This exponential functional form is justified by experimental studies on conducted vascular responses [41, 42]. The cumulative HPV signal is re-scaled by $T_{Max}$ and $T_{min}$ so that these parameters define the upper and lower bounds of the target vessel tone, $T_{j,\infty}$. The instantaneous tone of the $j$'th arterial segment is governed by first order kinetics

$$\tau \frac{dT_j}{dt} = T_{j,\infty} - T_j \qquad (25)$$

where $\tau$ is a time constant. Eq (25) operates as a linear filter to smoothly relax the instantaneous vessel tone, $T_j$, toward the target vessel tone, $T_{j,\infty}$. The vessel tone variable $T_j$ is assumed to govern arterial vessel diameter (resistance) and stiffness. This is incorporated into our lumped-parameter vascular mechanics framework by adjusting the resistances and compliances of each arterial vessel segment by

$$R_j = R_{0,j} T_j \text{ and,} \qquad (26)$$

$$C_j = \frac{C_{0,j}}{T_j}, \qquad (27)$$

where $R_j$ and $C_j$ are the resistance and compliance of the $j$'th segment, and $R_{0,j}$ and $C_{0,j}$ are the corresponding fixed nominal estimates of those parameters. The motivation behind the functional forms of Eqs (26) and (27) is that an increased vessel tone will increase vascular resistance and decrease the vessel compliance (increase stiffness) simultaneously. The vessel-wall resistance $R_D, j$ is computed by

$$R_D, j = \frac{\nu}{C_j}, \qquad (28)$$

where $\nu$ is the constant vessel relaxation speed and is recomputed whenever $C_j$ is altered by the action of HPV.

## Model parameterization

**Vascular network morphometry.** We use our distributive branching algorithm to generate a synthetic population of 3 rat lungs as shown in Fig 3 and S1 Table. Each vascular network is intended to represent a lobe of the rat lung. The shape of each mesh used to generate the vascular networks was chosen to emulate the shape of lungs lobes. We analyze the morphometry of these vascular networks with diameter-defined Strahler Ordering and compare our networks to the data from Jiang et al. [35], where they measured the length and radius of silicone-elastomer casts of rat pulmonary arterial vascular trees. Qualitatively, our networks capture the fractal-like branching pattern of true pulmonary arterial trees. To calibrate our model morphometry to the data in Jiang et al [35], we manually adjusted the hydraulic conductivity per unit length ($k$). Manual calibration required to find a satisfactory value for k was minimal, as we primarily varied it by orders of magnitude, and found $k = 10^{12} \, \mu m^2/mmHg/s$ to be in good agreement with the data from Jiang et al.

**Vascular mechanics.** To specify the vascular mechanics parameters ($R_0$, $C_0$, $R_D$, and $L$) for each arterial and venous vessel segment, we use the estimators defined in [38] (see Table 1). For $R_0$ we use Poiseuille's law shown in Eq (1) as the estimator. Treating vessels as composed

of a homogeneous linear elastic material,

$$C_0 = \frac{2\pi r^3 l}{hE}, \tag{29}$$

where $E$ is the Young's modulus and $h$ is the vessel thickness. We use an estimated value for $E$ from Blanco et al. [43]. Vessel wall thickness $h$ is calculated using the following relation:

$$h = r(ae^{br} + ce^{dr}), \tag{30}$$

where $a$, $b$, $c$, and $d$ are the fitting parameters also taken from Blanco et al. [43]. The vessel-wall resistance $R_D$ is estimated by

$$R_D = \frac{v}{C}, \tag{31}$$

where $v$ is the time constant for stress relaxation (see Table 1). The hydraulic intertance $L$ is estimated by

$$L = \frac{\rho}{\pi r^2}, \tag{32}$$

where $\rho$ is the blood density.

For the capillary segments, we set $R_0$ and $L$ to the same value as the upstream arterial segment, and use the estimator defined by Eq (31) for $R_D$. We set capillary compliance ($C$) by

$$C = \frac{V_{cap,j,un}}{\Delta P}, \tag{33}$$

where $V_{cap,j,un}$ is the unstressed blood volume of the $j$-th capillary segment, and $\Delta P$ is the areriolar-venular pressure drop. $\Delta P$ is set to 1 mmHg, $V_{cap,j,un}$ is defined by

$$V_{cap,j,un} = 0.7a_j V_{cap,Total}, \tag{34}$$

where $V_{cap,Total}$ is the total capillary blood volume, and $a_j$ is the fraction of area the corresponding perfusion zone occupies in the total area of the network domain. The 0.7 factor reflects the assumption that when the vessels are unstressed they contain 70% of the potential maximum total volume, as estimated by Gelman [49]. We assume that one half of the total pulmonary blood volume is in the capillaries, one quarter of the total pulmonary blood volume is in the arteries, and the other quarter is in the veins. Thus, $V_{cap,Total}$ is defined by

$$V_{cap,Total} = 2\pi \sum_{i=1}^{N} r_i^2 l_i, \tag{35}$$

where $N$ is number of arterial (or venular) segments in the network, and $r_i$ and $l_i$ are the radius and length of $i$-th segment. The summation in Eq (35) is multiplied by $\pi$ as we are computing the volume of blood in idealized cylindrical vessels, and is multiplied by 2 as we assume the venous network morphometry mirrors the arterial network.

To confirm that our model recapitulates realistic hemodynamics, we compare our simulations to data from isolated-ventilated-perfused rat lung experiments [20, 50]. In these experiments, the lungs and heart are excised *en bloc* from an animal. The left and right ventricle are punctured with cannulae so that the pulmonary circulation can be perfused, and the trachea is cannulated with a ventilator tube to control the airway pressure. The organ is kept in Zone 3 conditions (alveolar pressure is less than the venous pressure) and the whole organ flow rate is varied to measure the arterial-venous pressure-drop as an indicator of pulmonary vascular

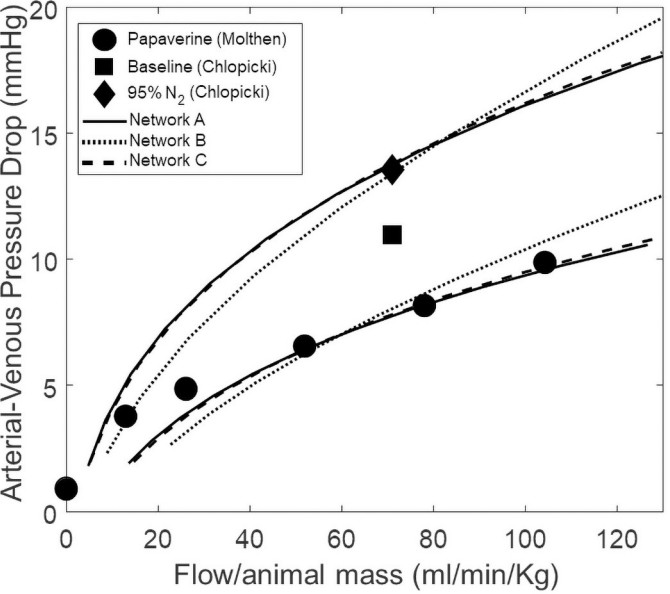

**Fig 4. Whole-organ pressure-flow relationships.** Discrete data points are taken from isolated-perfused-ventilated lung experiments performed in Molthen et al. [50] and Chlopicki et al. [20]. Simulations are run under Zone 3 conditions (venous outlet pressure (7 mmHg) > alveolar pressure (6 mmHg)). The model simulation curves that are near the 95% $N_2$ (Chlopicki) data point were run with $T^* = 0.8$, and the curves near the Papaverine (Molthen) data points were run with $T^* = 0.5$.

resistance. Molthen et al used papaverine during all of these experiments [50]. Thus data from this study are considered to represent a maximally dilated scenario. Chlopicki et al. investigated putative molecular mechanisms of HPV by applying a variety of pharmacological agents to a isolated-ventilated-perfused rat lungs while ventilated with normoxic (21% $O_2$, 5% $CO_2$, $N_2$ balance) and hypoxic (5% $CO_2$, 95% $N_2$) gaseous mixtures [20]. We have taken data points from this study where lungs were ventilated with normoxic and hypoxic gaseous mixtures without the presences of vasoactive pharmacological agents. The hypoxic condition is considered to represent a maximally vasoconstricted scenario, and the normoxic condition is considered a physiological baseline. Simulations run under conditions that emulate these experiments are compared to the data to determine upper and lower bounds for our model's pressure-flow relationships. We introduce a global vascular tone parameter $T^*$ (unitless) that is multiplied against the vascular resistance parameters ($R_0$) and divides the compliance parameters ($C_0$)—similar to Eqs (26) and (27). By manually adjusting the value of $T^*$ we are able to shift the pressure-flow relationship to a more constricted or dilated scenario. Fig 4 shows how our model compares to the data. The model simulation curves that are near the 95% $N_2$ (Chlopicki) data point were run with $T^* = 0.8$, and the curves near the Paperavine (Molthen) data points were run with $T^* = 0.5$.

**Oxygen transport and regulation by HPV.** To calibrate our oxygen transport and empirical model of HPV, we compare our simulations to human experiments done by Cheng et al, in which they measure pulmonary arterial pressure, ventilation, cardiac output, and systemic arterial oxygen partial pressure while inspiring normal (21% $O_2$, 5% $CO_2$) or hypoxic (10% $O_2$, 5% $CO_2$) air [47]. We re-scale the human valued cardiac output and ventilation measurements

to reflect that of an adult rat (350 g, 80 mL/min) so that

$$Q_P^* = Q_{P,\text{Cheng}} \frac{CO_{\text{Rat,ref}}}{CO_{\text{Human,ref}}} \quad \text{and,} \tag{36}$$

$$Q_V^* = Q_{V,\text{Cheng}} \frac{CO_{\text{Rat,ref}}}{CO_{\text{Human,ref}}}, \tag{37}$$

where $Q_P^*$ and $Q_V^*$ are the magnitude of total blood flow and ventilation through the model respectively, $Q_{P,\text{Cheng}}$ and $Q_{V,\text{Cheng}}$ are the cardiac output and ventilation data from Cheng et al respectively, $CO_{\text{Rat,ref}}$ is a reference value for the basal cardiac output of an adult rat, and $CO_{\text{Human,ref}}$ is a reference value for the basal cardiac output of an adult human (5 L/min). We note that the hypoxia exposure activates the chemoreflex and local metabolism-mediated vaso-dilation to increase cardiac output. Thus there is a change in the total flow-rate through the pulmonary circulation independent of the action of HPV or other oxygen-sensitive pulmonary vasoregulation.

To specify the oxygen sensitivity parameter, $P_{HPV}$, we consider the observation that major changes in mean pulmonary arterial pressure or total pulmonary resistance from HPV are generally only observed when alveolar oxygen falls below 80 mmHg [51]. With this in mind we want to ensure that the HPV signal ($S$) is equal to 1/2 when alveolar oxygen is 80 mmHg ($S$ is bounded by the interval (0, 1)). This is achieved by solving Eq (23) for $P_{HPV}$. Substituting $S = 1/2$ and $P_{alv,O2}$ 80 mmHg yields

$$P_{HPV} = \frac{-P_{alv,O2}}{\ln(S)} = \frac{80}{\ln(2)} = 115.42 \approx 115. \tag{38}$$

We use the cardiac output, ventilation, and pulmonary arterial pressure measurements from Cheng et al. as boundary conditions (data inputs) for our model simulations and compare the oxygen partial pressure in the primary vein of our model to the systemic arterial oxygen blood gas measurements from Cheng et al. [47]. The parameters representing apparent oxygen diffusivity ($D_{O2}$), maximum tone from HPV ($T_{Max}$), and minimum tone from HPV ($T_{min}$) are empirical quantities that are not able to be experimentally measured. We calibrate the model prediction of blood oxygenation by optimizing these parameters with the BELUGA Genetic Algorithm implemented in MATLAB [48]. This optimization scheme seeks to minimize the objective function

$$J = \sum_{i=1}^{2} (P_{O2,D,i} - P_{O2,M,i})^2, \tag{39}$$

where $P_{O2,D}$ and $P_{O2,M}$ denote the data from Cheng et al. and model prediction of oxygen tension in the primary vein respectively. $i = 1$ is the normoxia condition, and $i = 2$ is the hypoxia condition. The model is solved over 120 respiratory cycles and the primary vein oxygen tension is averaged over the last 2 respiratory cycles. We run BELUGA for 30 generations with 100 individuals and allow for mutation. (During these BELUGA runs we set $\tau = 2$ s so that the model reaches steady state in less simulation time. The operative condition is that $\tau > R_R$, so that the HPV model is able to average over at least one respiratory cycle.) Table 2 shows the values of our model predicted oxygen tension, final cost from Eq (39), and values of the optimized parameters. Fig 5 graphically shows how the model predicted oxygen tension compares to the data. We also report the standard deviation of each parameter estimate and correlations between them in Table 3. One can observe that $T_{Max}$ and $T_{min}$ are highly correlated for all

**Table 2. Results from BELUGA Genetic Algorithm.** The top block contains oxygen tensions from Cheng et al., and most fit individuals for each network from BELUGA optimization. The middle block contains the value of our objective function for each network from the most fit individuals. The bottom block contains the best fit parameter values for each network.

|  | Network A | Network B | Network C | Cheng et al. |
|---|---|---|---|---|
| $P_{O2,M}$ (FiO2 = 0.21) | 92.51 | 89.30 | 92.25 | 98.31 |
| $P_{O2,M}$ (FiO2 = 0.10) | 57.76 | 60.50 | 61.68 | 49.09 |
| $J$ | 104.92 | 196.97 | 159.64 |  |
| $D_{O2}$ | 3.36 | 14.60 | 0.95 |  |
| $T_{Max}$ | 1.49 | 1.48 | 1.49 |  |
| $T_{min}$ | 0.49 | 0.48 | 0.31 |  |

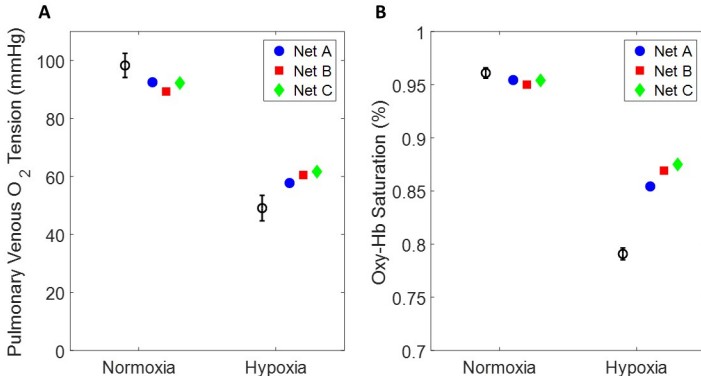

**Fig 5. Fits from BELUGA Genetic Algorithm.** (A) Oxygen partial pressure in the pulmonary vein. The block circle and error bars denote data from Cheng et al., the blue circle denotes network A, the red square denotes network B, and the green diamond denotes network C. The plotted point is the best fit individual from BELUGA (parameters reported in Table 2). (B) The same data and results from (A) but expressed in terms of oxyhemoglobin saturation.

three networks. This is an expected result, and these two parameters are directly subtracted from each other in Eq (24) of the empirical HPV model.

**Sensitivity analysis.**    To assess the sensitivity of model prediction on the variation in model parameters we perform a sensitivity analysis on the parameters listed in Table 4. We exclude parameters from this analysis that were involved in network generation or specific to each vessel segment to focus on parameters that influence system level function. Some parameters that have been directly measured have been excluded ($\alpha$—solubility of oxygen in plasma, $\beta$—solubility of oxygen in air, $\rho$—density of blood, $C_{Hb}$—oxyhemoglobin binding site concentration). We vary the baseline of each parameter ±10%. Our sensitivity index is defined as

$$\mathcal{S}_i = 100 \times \left( \frac{P_{O2,i} - P_{O2,ref}}{P_{O2,ref}} \right), \tag{40}$$

**Table 3. Standard Deviation (SD) and Correlations between parameters estimated by BELUGA Genetic Algorithm.**

| Network A |  |  |  | Network B |  |  |  | Network C |  |  |  |
|---|---|---|---|---|---|---|---|---|---|---|---|
|  | $D_{O2}$ | $T_{Max}$ | $T_{min}$ |  | $D_{O2}$ | $T_{Max}$ | $T_{min}$ |  | $D_{O2}$ | $T_{Max}$ | $T_{min}$ |
| SD | 1,025.9 | 13.52 | 4.22 | SD | 1,952.4 | 12.80 | 4.08 | SD | 1,213.1 | 13.66 | 3.04 |
| $D_{O2}$ | 1 | 0.20 | 0.19 | $D_{O2}$ | 1 | 0.41 | 0.45 | $D_{O2}$ | 1 | 0.20 | 0.19 |
| $T_{Max}$ | - | 1 | 0.96 | $T_{Max}$ | - | 1 | 0.96 | $T_{Max}$ | - | 1 | 0.97 |
| $T_{min}$ | - | - | 1 | $T_{min}$ | - | - | 1 | $T_{min}$ | - | - | 1 |

**Table 4. Sensitivity analysis.** Percent change in oxygen tension of the primary vein given a ±10% perturbation in nominal parameter value. $^*$—The sensitivity index is less than 0.01.

| Parameter | | | Network A | Network B | Network C |
|---|---|---|---|---|---|
| $E$ | +10% | | 5.79 | 5.06 | 4.12 |
| | −10% | | −6.40 | −4.92 | −4.06 |
| $v$ | +10% | | −0.12 | −0.02 | 0.11 |
| | −10% | | −0.28 | 0.00 | −0.08 |
| Hct | +10% | | −11.40 | −11.26 | −10.55 |
| | −10% | | 16.10 | 17.46 | 16.33 |
| $n$ | +10% | | −3.39 | −3.59 | −3.43 |
| | −10% | | 3.44 | 4.50 | 4.17 |
| $P_{50}$ | +10% | | 1.51 | 2.28 | 2.80 |
| | −10% | | −1.94 | −2.54 | −2.62 |
| $C_{O2,input}$ | +10% | | 8.70 | 9.31 | 9.55 |
| | −10% | | −7.57 | −7.81 | −7.71 |
| $R_R$ | +10% | | −0.20 | −0.30 | −0.47 |
| | −10% | | 0.87 | 0.53 | 0.84 |
| $Q_V^*$ | +10% | | 2.88 | 2.07 | 2.02 |
| | −10% | | −3.11 | −2.09 | −1.85 |
| $P_{alv}^*$ | +10% | | −0.32 | 0.14 | 0.00$^*$ |
| | −10% | | −0.21 | 0.26 | 0.42 |
| $P_{HPV}$ | +10% | | 1.97 | 1.83 | 1.54 |
| | −10% | | −2.66 | −2.09 | −1.60 |
| $\lambda$ | +10% | | 0.70 | 0.25 | 0.30 |
| | −10% | | −1.28 | −0.29 | −0.25 |
| $D_{O2}$ | +10% | | −0.20 | −0.01 | 0.03 |
| | −10% | | −0.28 | −0.08 | −0.04 |
| $T_{Max}$ | +10% | | 6.28 | 6.33 | 5.26 |
| | −10% | | −6.41 | −5.60 | −4.83 |
| $T_{min}$ | +10% | | 2.81 | 1.53 | 1.16 |
| | −10% | | −3.09 | −1.52 | −1.37 |

where $\mathcal{S}_i$ is the sensitivity of the $i$-th parameter of interest (percent change in oxygen leaving the pulmonary circulation), and $P_{O2,ref}$ and $P_{O2,i}$ are the model simulated oxygen tensions in the primary vein run with all parameters fixed at their nominal values and the $i$-th parameter perturbed, respectively. The nominal parameter values are listed in Table 1, and we treat the BELUGA estimated values for $D_{O2}$, $T_{Max}$, and $T_{min}$ as their nominal values.

Results of our sensitivity analysis are shown in Table 4. The most sensitive parameters are hematocrit (Hct), where a 10% perturbation results in a $\sim 13\%$ change in $\mathcal{S}$, and the oxygen concentration entering the pulmonary circulation ($C_{O2,input}$), where a 10% perturbation results in a $\sim 8\%$ change in $\mathcal{S}$. Of the BELUGA estimated parameters, $T_{Max}$ is the most sensitive ($\mathcal{S} \approx 5.5$), $T_{min}$ is mildly sensitive ($\mathcal{S} \approx 1.5$), and $D_{O2}$ is relatively insensitive ($\mathcal{S} \approx 0.1$).

## Results

### Effect of HPV on regional hemodynamics

To determine the effect of HPV on flow distribution, we run simulations with and without our empirical HPV model. We also simulate the effect of uniform vasoconstriction (UVC) by setting $T^* = 0.8$ as a control. These simulations were run with an artificial condition of uniform

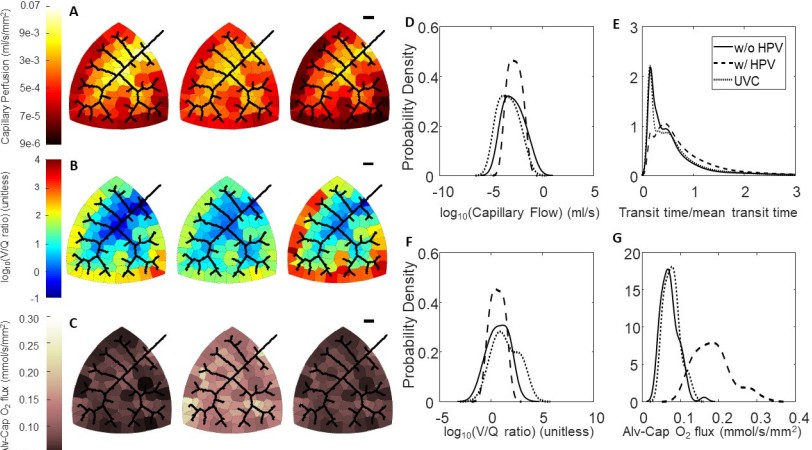

**Fig 6. Effect of HPV and uniform vasoconstriction on Network A regional perfusion, V/Q matching, and oxygen flux.** (A-C) Leftmost column of networks are without regulation from HPV, middle column of networks are with regulation from HPV, and the rightmost column of networks are with uniform vasoconstriction, where $T^* = 0.8$ is used for the global vascular tone. The black scale bar is 1000 $\mu$m. (A) Perfusion in the capillary compartments; (B) Ventilation-perfusion (V/Q) ratios in the capillary compartments; (C) Oxygen flux in the capillary compartments. (D-G) Blood flow, RBC transit times, V/Q ratios, and oxygen flux expressed as probability densities—calculated with a kernel density estimator. (D) Distribution of flow; (E) RBC transit times normalized to mean transit time; (F) Distribution of V/Q ratios; (G) Distribution of oxygen flux.

airflow distribution (each alveolar compartment receives airflow that is proportional to its size). In the absence of regulation by HPV, our model predicts that shorter path lengths tend to have greater flow compared to longer path lengths as shown in Figs 6A, 7A and 8A. However, in the presence of regulation by HPV one can appreciate that there is a homogenization of blood flow. UVC reduces the total flow through the network similar to HPV, however the

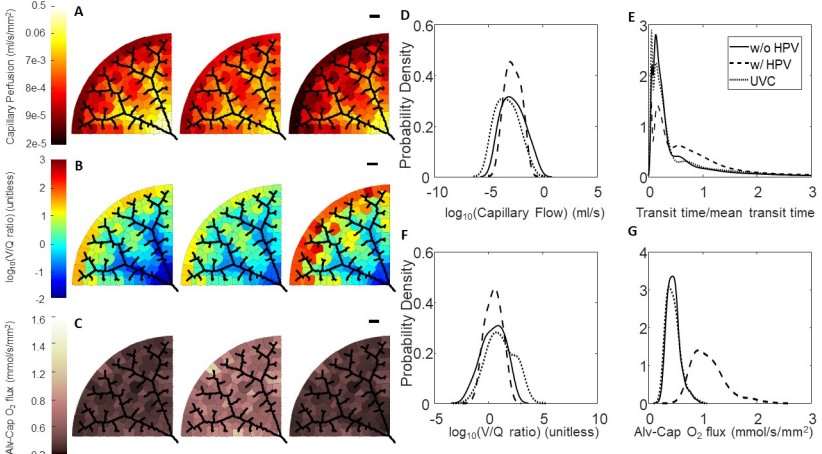

**Fig 7. Effect of HPV and uniform vasoconstriction on Network B regional perfusion, V/Q matching, and oxygen flux.** (A-C) Leftmost column of networks are without regulation from HPV, middle column of networks are with regulation from HPV, and the rightmost column of networks are with uniform vasoconstriction, where $T^* = 0.8$ is used for the global vascular tone. The black scale bar is 1000 $\mu$m. (A) Perfusion in the capillary compartments; (B) Ventilation-perfusion (V/Q) ratios in the capillary compartments; (C) Oxygen flux in the capillary compartments. (D-G) Blood flow, RBC transit times, V/Q ratios, and oxygen flux expressed as probability densities—calculated with a kernel density estimator. (D) Distribution of flow; (E) RBC transit times normalized to mean transit time; (F) Distribution of V/Q ratios; (G) Distribution of oxygen flux.

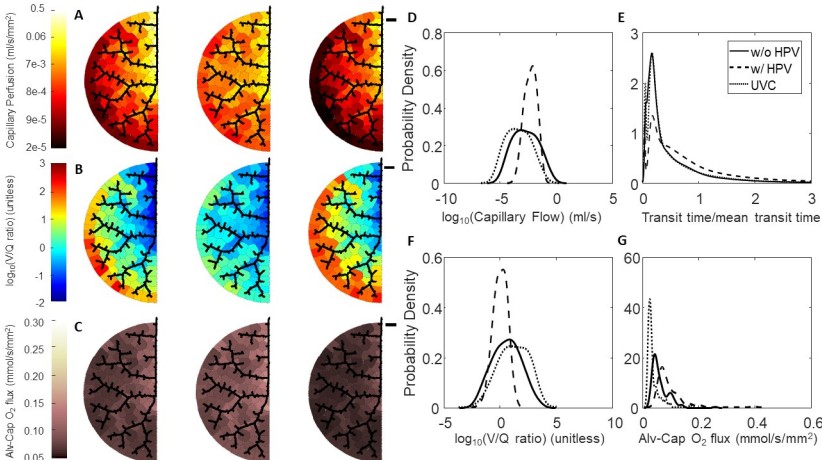

**Fig 8. Effect of HPV and uniform vasoconstriction on Network C regional perfusion, V/Q matching, and oxygen flux.** (A-C) Leftmost column of networks are without regulation from HPV, middle column of networks are with regulation from HPV, and the rightmost column of networks are with uniform vasoconstriction, where $T^* = 0.8$ is used for the global vascular tone. The black scale bar is 1000 $\mu$m. (A) Perfusion in the capillary compartments; (B) Ventilation-perfusion (V/Q) ratios in the capillary compartments; (C) Oxygen flux in the capillary compartments. (D-G) Blood flow, RBC transit times, V/Q ratios, and oxygen flux expressed as probability densities—calculated with a kernel density estimator. (D) Distribution of flow; (E) RBC transit times normalized to mean transit time; (F) Distribution of V/Q ratios; (G) Distribution of oxygen flux.

distribution of flow is very similar to the simulation run without regulation from HPV but slightly shifted to lower flows. This is graphically illustrated in Figs 6D, 7D and 8D and can also be understood by observing that the coefficient of variation for capillary blood flow decreases for all three networks (Table 5).

Another way to interrogate the hemodynamic effects of HPV is to observe the red blood cell (RBC) transit time distribution. Experimentally, the RBC transit time distribution can be measured via an indicator-dilution experiment. In these experiments a vascular bed with a defined single inlet and outlet is identified. Then the inlet is then perfused with a bolus of tracer dye (ex: Evan's Blue, FITC conjugated dextran), and then the washout is measured at the outlet [52]. The time series of tracer dye concentration at the inlet, $C_{in}(t)$, and

**Table 5. Effect of HPV on regional flow, V/Q matching, and oxygenation.** Model simulations were performed with uniform vasoconstriction (UVC), and with and without regulation by HPV. All simulations used a fixed 25 mmHg arterial-venous pressure drop. The top block summarizes how HPV and uniform vasoconstriction (UVC) affects the total flow through the organ, and the variability of blood flow, V/Q ratio by the Coefficient of Variation (CV = $\mu/\sigma$). The bottom block summarizes how HPV impacts the oxygenation in terms of the total alveolar-capillary oxygen mass flux, oxygen mass flux CV, oxygen mass flux normalized to total oxygen flux, venous oxygen tension, and venous oxyhemoglobin saturation.

| | Network A | | | Network B | | | Network C | | |
|---|---|---|---|---|---|---|---|---|---|
| | w/o HPV | w/ HPV | UVC | w/o HPV | w/ HPV | UVC | w/o HPV | w/ HPV | UVC |
| Total Flow (ml/s) | 1.10 | 0.38 | 0.26 | 2.32 | 0.62 | 0.60 | 1.77 | 1.07 | 0.46 |
| Total Resistance (mmHg s/ml) | 28.03 | 80.13 | 117.64 | 13.32 | 49.45 | 51.16 | 17.47 | 28.78 | 67.13 |
| Flow CV | 2.47 | 1.28 | 2.42 | 2.35 | 1.42 | 2.44 | 2.09 | 1.08 | 2.35 |
| V/Q CV | 3.42 | 1.24 | 2.45 | 1.78 | 1.65 | 2.75 | 3.39 | 1.36 | 1.65 |
| Total Alv-cap $O_2$ flux (mmol/s) | 4.6 | 11.6 | 5.0 | 29.7 | 70.1 | 28.8 | 3.8 | 5.6 | 2.3 |
| Alv-cap $O_2$ flux CV | 0.30 | 0.25 | 0.26 | 0.25 | 0.27 | 0.28 | 0.52 | 0.48 | 0.58 |
| $O_2$ flux CV / Total $O_2$ flux (s/mol) | 66.19 | 22.14 | 51.9 | 8.37 | 3.84 | 9.71 | 139.19 | 87.23 | 256.33 |
| Venous $O_2$ (mmHg) | 81.26 | 100.20 | 78.03 | 56.66 | 96.45 | 61.72 | 54.79 | 74.63 | 60.57 |
| Venous $O_2$ (%) | 0.94 | 0.97 | 0.93 | 0.85 | 0.96 | 0.87 | 0.84 | 0.92 | 0.87 |

concentration at the outlet, $C_{out}(t)$, are related by

$$C_{out}(t) = C_{in}(t) * h(t) = \int_0^\infty C_{in}(s)h(t-s)ds, \tag{41}$$

where $*$ is the convolution operator, and $h(t)$ is a probability distribution of times it could take for a RBC to traverse the vascular bed. Figs 6E, 7E and 8E show the RBC transit time distributions for each of our model networks. In the absence of regulation by HPV our model networks have some paths that are highly perfused and other paths that are barely perfused at all. In Figs 6E, 7E, and 8E we show how the action of HPV homogenizes $h(t)$ and increases the mean transit time for all three of our vascular networks. We emphasize that an increased transit time implies that a RBC will have more opportunity to acquire oxygen as it traverses the pulmonary circulation.

## Effect of HPV on V/Q matching and regional alveolar-capillary oxygen flux

The consequence of homogenization of capillary blood flow and RBC transit times is that V/Q matching becomes more homogeneous as the whole organ coefficient of variation decreases for all three networks, as shown in Table 5 and illustrated in Figs 6B and 6F, 7B and 7F and 8B and 8F. UVC shifts the V/Q ratio to the right, which reflects that there are more perfusion zones with large V/Q ratios (due to low blood flow) under this condition. Moreover, regional alveolar-capillary oxygen flux is also more evenly distributed throughout the organ with physiological control from HPV. This is graphically shown in Figs 6C and 6G, 7C and 7G and 8C and 8G and can also be understood by observing that the coefficient of variation for regional alveolar-capillary flux decreases for all three networks with the action of HPV (see Table 5). Perhaps more importantly, the presence of HPV increases the whole-organ oxygen transport into the bloodstream, and the end total amount of oxygen leaving the pulmonary circulation. UVC has variable effects on the distribution of alveolar-capillary oxygen flux, and does increase the amount of oxygen entering the bloodstream—although not as much as HPV. These results are summarized in Table 5.

## Effect of the fraction of inspired oxygen on HPV-mediated redistributions in hemodynamics, V/Q matching, and alveolar-capillary oxygen flux

Simulations illustrated in Figs 6, 7 and 8 are run under normoxic conditions—the fraction of inspired oxygen ($F_{i,O_2}$) is set to 21%, which corresponds to a partial pressure of 150 mmHg. To determine how the $F_{i,O_2}$ influences HPV-mediated redistributions in blood flow, V/Q ratios, and oxygen flux, we performed a series of simulations where we set $F_{i,O_2}$ to 10%, 15%, 21%, and 50%. The results of these simulations for Network A are shown in Fig 9 (results from the Networks B and C are similar and not shown). These simulations predict that the distribution of blood flow, V/Q ratios, and RBC transit time is homogenized with decreasing fractions of $F_{i,O_2}$ (Fig 9A, 9B, 9D and 9E). A larger $F_{i,O_2}$ flattens the distribution of oxygen flux (Fig 9C). This is reflected by the decreased CV for $F_{i,O_2}$ below 21%—indicating that the distribution is being homogenized as $F_{i,O_2}$ gets smaller (Fig 9E). Moreover, the mean oxygen flux decreases (leftward shift in the distribution) with a smaller $F_{i,O_2}$.

## Effect of airway occlusions on HPV-mediated redistributions in hemodynamics, V/Q matching, and regional alveolar-capillary oxygen flux

HPV is understood to direct blood flow away from regions of the lung that are under venti-lated and have a poor oxygen supply. We simulate this behavior by setting alveolar airflows $q_i$ = 0. For these simulations we hold the bulk airflow ($Q_V^*$) constant, so that occluding an single

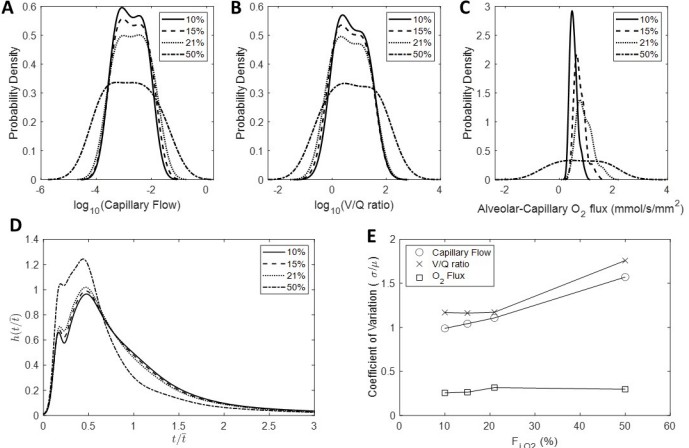

**Fig 9. Effect of the fraction of inspired oxygen on HPV-mediated redistributions in hemodynamics, V/Q matching, and oxygen flux.** (A) Distributions of capillary blood flow; (B) Distributions of V/Q ratios; (C) Distributions of oxygen flux; (D) RBC transit time distributions; (E) Coefficient of variation of blood flow, V/Q ratio, and oxygen flux plotted as a function $F_{i,O2}$.

alveolus will result in an increased airflow to the unoccluded alveoli. We occlude the airway of the most southwestern perfusion zone, and progressively occlude more perfusion zones following the structure of the vascular network. The results of these simulations for Network A are shown in Fig 10 (results from Networks B and C are similar and not shown). These simulations predict that HPV directs blood flow away from the regions with occluded alveoli (Fig 10A–10C). As the fraction of occluded alveoli increases there is a gradual decrease in the end-venous oxygen tension (Fig 10D); a gradual increase in the Pearson correlation between

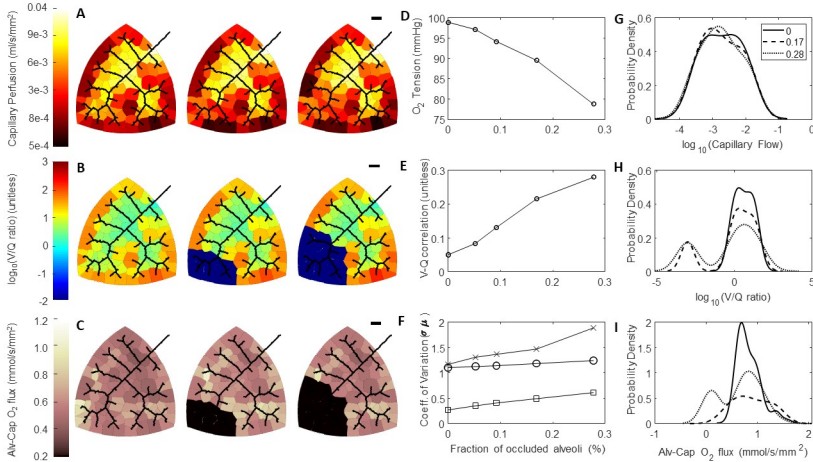

**Fig 10. Effect of airway occlusions on HPV-mediated redistributions in hemodynamics, V/Q matching, and oxygen flux.** (A-C) Leftmost column of networks are with 0% of the alveoli occluded, middle column of networks are with 17% of the alveoli occluded, and the rightmost column of networks are with 28% of the alveoli occluded. The first airway occlusion is at the most southwestern perfusion zone, and progressive airway occlusions are made following the structure of the vascular network. The black scale bar is 1000 $\mu$m. (A) Perfusion in the capillary compartment; (B) Ventilation-Perfusion (V/Q) ratios in the capillary compartment; (C) Oxygen flux in the capillary compartments; (D) End-venous oxygen tension as a function of the fraction of occluded alveoli; (E); Pearson correlation between ventilation and perfusion as a function of the fraction of occluded alveoli; (F) Coefficient of Variation for blood flow (circles), V/Q ratios (x's), and oxygen flux (squares); (G) Distribution of blood flow; (H) Distribution of V/Q ratios; (G) Distribution of oxygen flux.

regional ventilation and regional blood flow (Fig 10E), and an increase the heterogeneity of V/Q ratios and regional oxygen flux (Fig 10F). Alveolar occlusions produce bimodal V/Q ratio and oxygen flux probability densities (Fig 10H and 10I), however the changes to the blood flow distribution are mild in comparison. This is also reflected by the small gradual increase in the blood flow coefficient of variation shown in Fig 10F. When we run these simulations without HPV, there is no change in the blood flow distribution in response to the airway occlusions (simulations not shown). Moreover, there is decrease in the end-venous oxygenation compared to simulations with HPV—for Network A there is approximately 78 and 64 mmHg of oxygen with 28% of the alveoli occluded with and without HPV respectively.

## Discussion

We constructed an integrated model of distributed gas exchange and regional blood flow control in lung tissue to investigate mechanisms regulating V/Q matching. In an isolated gas exchange unit, increasing the upstream vascular resistance would decrease blood flow, reduce RBC transit time, and thus improve oxygenation. Yet a change in flow to a single unit will impact other gas exchange units in the network. Regional flows and gas exchange throughout the lung are coupled in a complex nonlinear manner. The overarching goal of this study was to gain insight into how this distributed multi-scale system works.

Specifically, this study presents the derivation of, and predictions from a multi-scale multi-physics model of ventilation-perfusion matching. This model consists of morphometrically realistic vascular networks, vascular fluid mechanics, oxygen transport, and an empirical model of hypoxic pulmonary vasoconstriction. The HPV model integrates flow regulation with oxygen transport in a physiologically realistic manner, and facilitates simulations and functional predictions of how active regulation from HPV affects blood flow and oxygen transport at the local and system level.

### Vascular network generation

Our novel distributive branching algorithm provides an efficient way to generate a vascular network over a 2D domain of any shape. While the present study is focused on lungs, this algorithm could be applied to generate vascular networks in any tissue of interest. Another potential use of this algorithm would be to impute missing data from vascular networks acquired from segmenting images from CT, MRI, or other radiological scans. One limitation of our generated networks is that they are two dimensional, whereas true vascular networks exist in 3 dimensional space. We chose to work with 2D domains to reduce the computational cost and simplify the interpretation of spatial results. However the operating principle of this algorithm can be easily extended to three dimensions. In future work we plan to generate 3D networks with this approach and compare/contrast them with true animal and human vascular geometries.

Another limitation of our network generation algorithm is our assumption of a constant hydraulic conductivity per unit length, in the form of the adjustable parameter $k$. This assumption was used to assign a radius to each vessel segment, where Poiseuille's law to was used impose conservation of flow as the operative constraint. The parameter $k$ was treated as an adjustable parameter. While this approach results in morphometrically realistic networks, how we assign radii to vessel segments could potentially be improved by using a different constraint other than Poiseulle's law. Murray's law, and related derivatives, represent potential alternative choices [53, 54].

## Modeling of vascular mechanics

We simulate hemodynamics using a lumped-parameter model inspired by the bond graph approach of Safaei et al [38]. We have extended Safaei's vessel segment model by adding a variable pressure source (represents the mechanical pressure of the airway), and deriving and implementing the flow and pressure balance boundary conditions for a venous network (when flows from two segments become confluent in a larger segment). Our primary interest was in representing the spatial distribution of blood flow in a deterministic manner. The model presented in this study is capable of capturing both the whole-organ mechanics as shown in Fig 4, and the regional distribution of flow as shown in Figs 6A and 6D, 7A and 7D and 8A and 8D. In Fig 4, one can observe that Network B does not fit the data as well as Network's A and C, as it appears to be mechanically stiffer. This is likely a result of how we assign radii to vessel segments in the network generation step, as discussed previously. There is a major bifurcation close to root of Network B, which results in there being effectively two parallel networks.

In real lungs, the CV of blood flow is dependent upon the size of the region observed. This is due to the fractal properties of blood vessel branching and blood flow in the pulmonary circulation. Glenny et al. [55] report a blood flow CV of 0.35 when sampling 0.53 mm$^3$ volumes of rat lung and a fractal dimension of 1.12. The blood flow CV we report from simulations without regulation from HPV are approximately 2.2, and with regulation from HPV the CV becomes approximately 1.2. On average, our perfusion zones are 750 $\mu$m$^2$ ($0.75 \times 10^{-4}$ mm$^2$). The linear dimension of an average perfusion from our model is $\sqrt{0.75 \times 10^{-4} \text{mm}^2} = 0.027$ mm, and we could extrapolate the volume to be $(0.027 \text{mm})^3 = 2.05 \times 10^{-5}$ mm$^3$. Using the fractal relationship reported in Glenny et al., we can extrapolate that a sampling volume of $2.05 \times 10^{-5}$ mm$^3$ corresponds to a CV of 1.01. Our predicted blood flow distributions are thus in the realm of reality, but do appear to overestimate the heterogeneity. This is likely due to the relatively small number of simulated perfusion zones—there are approximately 100 perfusion zones in each network, whereas Glenny et al. used 1000's of sampling volumes when analyzing blood flow distribution in the entire rat lung.

It has been observed that there is an uneven partitioning of hematocrit that is discharged into daughter vessels at bifurcations [56]. This phenomena has implications for both oxygen transport, and rheology. We used to Preis et al.'s non-linear viscosity law as a compromise between accounting rheological complexities and minimizing computational cost [36]. This empirical viscosity law enables one to compute viscosity as a function of vessel segment radius and hematocrit. However, our implementation leverages the apparent viscosity's dependence on radius while assuming that hematocrit is constant in every vessel segment. To our knowledge, no study has measured the uneven partitioning of hematocrit in the pulmonary circulation, so we do not have adequate data so specify a model that captures this phenomena. It is plausible to consider that the uneven distribution of red blood cells would lead to a heterogeneous distribution of apparent viscosities throughout the pulmonary vasculature and that could have major consequences on the lungs ability to regulate blood flow distribution.

## Modeling of oxygen transport

Our oxygen transport model was derived on the basis of compartmental mass-balance. A variety of models have been derived to capture the mechanics and transport properties in sheet flow [39, 40]. Given that smallest vessel segments in our model are not the size of pulmonary capillaries, we reasoned that using a sheet flow model would add unnecessary computational expense. Instead, we treat a single capillary vessel segment interacting with an alveolar compartment as the gas exchange unit. We justify this choice by representing the diffusion of oxygen across the alveolar-capillary boundary with an apparent oxygen diffusivity parameter,

$D_{O2}$. The function of sheet flow is that it increases the surface area of the blood-air interface. We can interpret increasing $D_{O2}$ as increasing the surface area of the blood-air interface.

In Fig 5 and Table 2, we can observe that the model struggles to capture the hypoxic condition. This is a result of two interacting phenomena. (1) We do not account for physiological changes in systemic oxygen consumption under hypoxic conditions. In other words, we assume that the oxygen concentration entering the pulmonary circulation is the same under normoxic and hypoxic conditions, as the model does not have a closed-loop of circulation. (2) How we account for oxygen bound the hemoglobin and freely dissolved in plasma. Even under hypoxic conditions, the model is still operating on the relatively plateaued portion on the oxyhemoglobin dissociation curve above the half-max saturation. In this region of the curve, it takes a large change in oxygen concentration to see a change in oxygen tension. The model fits to the hypoxic conditions could potentially be improved by accounting for the closed-loop circulation and a physiological representation of oxygen consumption.

## Empirical modeling of hypoxic pulmonary vasoconstriction

There is currently no consensus in the field regarding the governing molecular pathways which underlie hypoxic pulmonary vasoconstriction. Our model is inspired by the conducted response pathway proposed by Keubler and colleagues. Conducted vascular responses offer a mechanistic explanation of how information regarding oxygen content in the alveolar space is transduced to the pulmonary arteries which control the distribution of blood flow. Wang et al. [22] demonstrates that the connexin-40 hemmichannel gap junction is essential for HPV in mice. Genetic deletion of connexin-40 abrogated the depolarization of the endothelium in the pulmonary arterioles and vasoconstriction when the mice were exposed to hypoxia. Based on their experiments they conclude that conducted vascular responses from the alveolar-capillary boundary to the upstream pulmonary arteries mediate HPV. Follow up work from this group claims that the upstream oxygen sensor is an oxygen-sensitive sphingomyelinase which ultimately activates the CFTR to act as a chaperone to ferry TRPC6 calcium channels to the membrane of vascular smooth muscle cells near the alveolar-capillary boundary [21].

If we accept the assumption that conducted vascular responses do indeed mediate HPV, then it would follow that any oxygen-sensitive pathway that results in the depolarization of the endothelium at or near the alveolar-capillary boundary could contribute to activation of HPV. Coincidentally, recent work by Duham et al. has proposed a pathway for HPV in which hydrogen peroxide generated by nicotinimide adenine dinucleotide dehydrogenase iron-sulfur protein 2 (Ndufs2) regulates the activity of Kv1.5 ion channels on the pulmonary arterial smooth muscle cells [23]. Given that Ndufs2 is a core subunit of mitochondrial complex 1, it is intuitive to expect the redox processes which mitochondria facilitate would be involved in oxygen-sensitive vasoregulation.

The empirical model of HPV derived in this study is a minimal representation of the conducted vascular response pathway. A more detailed model of conducted vascular responses would represent the electrophysiology of the blood vessel wall (membrane potential, intracellular calcium concentration of pulmonary arterial smooth muscle cells, etc). Simulations with our model predict that under an artificial test condition of having a uniform airflow distribution, the activation of HPV matches perfusion to ventilation. Under this test condition, the distribution of blood flow is homogenized. The critical prediction from our empirical HPV model is that homogenization of blood flow and V/Q ratio distributions improves oxygen transport into the bloodstream. We report this in terms of the homogenization of regional alveolar-capillary oxygen flux, increased whole-organ alveolar-capillary oxygen mass transport, and increased oxygen within the primary vein (Figs 6–9 and Table 5). A caveat on to

these interpretations is our use of a fixed arterial-venous pressure drop of 25 mmHg. In reality vascular pressure, flow, and resistance are all dynamic quantities influenced by a number of pathophysiological systems. Using a fixed arterial-venous pressure drop allows us to assess how physiological regulation of vascular resistance by HPV affects both total blood flow through the pulmonary circulation and the regional distribution of flow throughout the network. We emphasize that in Table 5 the total flow and vascular resistance are similar between the HPV and uniform vasoconstriction (UVC) simulations, while HPV does a much better job at improving oxygenation compared to UVC. The differences in the ability of HPV and UVC to affect oxygen uptake is explained by how they alter the distribution of blood flow and V/Q matching.

Measurements of pulmonary flow distribution in pigs have been performed by Hlasta et al. and Starr et al. using a microsphere infusion technique [14, 15]. They observe that the HPV response is spatially heterogeneous, and conclude that this heterogeneity may be related to baseline heterogeneity of the airflow distribution. We performed a series of simulations where we occluded alveolar compartments to explore the effects of a non-uniform airflow distribution (Fig 10). We observe that our model is indeed able to divert blood flow away from regions with occluded alveoli and limited oxygen supply. However, these simulations are still limited. One limitation is that we do not have an accurate representation of realistic and heterogeneous airflow distributions. The homogenization of blood flow by HPV that our model predicts is a consequence of the uniform air flow distribution we use. The heterogeneity of real pulmonary airflow distributions are almost certainly related to the anatomical structure of the airway. Interestingly, in a healthy individual airflow and blood flow are both heterogeneously distributed yet the V/Q ratios are still well matched. There is currently no uniformly accepted explanation of why V/Q ratios are well matched, but it is likely combination of the intervowen nature of airway and vascular anatomies, passive phenomena such as the influence of gravity, and the physiological regulatory mechanisms (such as HPV). Future model development will need to integrate our model of vascular flow regulation with a model of airflow through a realistic network structure to systematically evaluate the contribution of regulation by HPV and anatomical geometry to basal V/Q matching. A second limitation is that the simulated diversion of blood flow away from regions with occluded alveoli is milder than we expect to observe in a real a true lung. This is most likely a limitation of our assumption that the vasoconstrictive ability of every vessel segment is bounded by the same maximum and minimum tone ($T_{Max}$ and $T_{min}$). S4 Fig shows how the tone from HPV in each vessel segment corresponding to the simulations in Fig 10. One can appreciate the largest tones tend to be in the distal segments and near regions with zero airflow, whereas the more proximal segments tend to have lower tone. This is explained by the mathematical form (exponential decay) of how we assume these signals are conducted upstream through vessel walls. An improved version of the model may require tone parameters that are unique to each size (ex: Strahler order) of vessel segment, or represents the electrophysiology of pulmonary endothelium and vascular smooth muscle cells.

To our knowledge, this is the first study to explore how the integrated action of HPV regulates V/Q matching and oxygen transport throughout the lungs. A limitation of the predictions we have presented is that we have only accounted for regulation by HPV. We have not represented the central and peripheral chemoreflexes which alter the cardiac output and rate and depth of ventilation in response to changes in circulating and oxygen and carbon dioxide (plasma acid-base balance). Moreover, we have not accounted for the influence of gravity on the distribution of blood flow and V/Q ratios. The importance of posture on pulmonary blood flow distribution and V/Q matching is well appreciated. This knowledge has been leveraged in clinical settings to improve V/Q matching and oxygen uptake in patients by moving them into

a prone position [9]. Integrating these mechanisms into our model would provide a framework to understand the whole-body consequences of and influences on V/Q (mis)matching.

## Model paramterization

The model presented in this study is fairly complex and is comprised of many parameters. When possible, we set parameters to values that have been measured and reported in the literature. Some parameters however, are empirical in nature and are not directly measureable. In particular, the maximum and minimum tone ($T_{Max}$ and $T_{min}$) in our HPV model, and the apparent oxygen diffusivity ($D_{O2}$). Thus we set values for these parameters by estimating them. Algebraically, $T_{Max}$ and $T_{min}$ are subtracted from one another in the model. From the stand point of structural identifiability, estimating both of these parameters bodes for a ill-conditioned optimization problem. Given that there are no data available to inform how we set these parameters *a priori*, this becomes a circular problem of not knowing what value to keep one parameter fixed at while we estimate the other. This issue is what ultimately led us to use BELUGA, a genetic algorithm. Using BELUGA enabled us optimize both of these parameters. This in contrast to gradient-based methods (ex: Levenberg-Marquardt) which are much more susceptible to numerical instability and/or non-reproducible results from parameter unidentifiability [57]. After estimating, $T_{Max}$, $T_{min}$, and $D_{O2}$ we performed a retrospective sensitivity analysis to understand how these and the other model parameters influence the ability of our model to simulate oxygen uptake. We found that hematocrit (Hct) and the oxygen concentration entering the pulmonary circulation ($C_{O2,input}$) were the most sensitive model parameters. Interestingly, increasing Hct decreased the end-venous oxygen tension and decreasing Hct increased the end-venous oxygen tension. While this may appear paradoxical, perturbing Hct did not dramatically change the oxygen concentration in the blood, rather it shifted the blood oxygen concentration-partial pressure relationship. In other words, an increase in Hct will result in a specified blood oxygen concentration corresponding to a smaller partial pressure. The other interesting result that emerged from our sensitivity analysis was that $D_{O2}$ is a relatively insensitive parameter. We interpret this result to indicate that our model's ability to uptake oxygen is primarily flow-limited, rather than diffusion-limited. This reinforces the importance of how blood flow distribution is regulated, as altering flow in one segment of the vascular network affects the flow in all other segments of the network.

## Vascular recruitment

Krogh first demonstrated the phenomenon of pulmonary capillary recruitment, whereas total blood flow to the lungs increases, previously unperfused capillaries open up and become perfused [58–60]. The net effect is that the average RBC transit time tends to stay the same over a range of cardiac outputs. This enables the blood in the pulmonary circulation to be sufficiently oxygenated despite the increased cardiac output and oxygen demand from the systemic tissues. In our model, increases in perfusion pressure and flow lead to increases in microvascular volume via the lumped pressure-volume relationship (6). The limitation of this representation of vascular recruitment is that it does not directly represent vascular networks that are collapsed/non-perfused and then become recruited. In Fig 10D, we can see that the end-venous oxygenation decreases as the fraction of occluded alveoli is increased. While this result intuitively makes sense, it is reasonable to expect a true lung would be more resilient to this type of challenge due the mechanism of capillary recruitment. A downside to function enabled by capillary recruitment is that it could be a major contributor to challenges in diagnosing chronic pulmonary embolism (PE). Hypoxemia is uncommon in chronic PE and is often identified during a diagnostic work-up in patients with unexplained pulmonary arterial hypertension [61]. A

future version of the model presented in this study will require a biophysical representation of capillary recruitment to capture to nuances of these disease states.

## Summary of main findings

In conclusion, we have developed an integrative computational model of V/Q matching mechanics, oxygen transport, and the physiological mechanism of HPV. Our model simulations predict that under the artificial test condition of a uniform ventilation distribution (1) HPV matches perfusion to ventilation; (2) HPV homogenizes regional alveolar-capillary oxygen flux; and (3) HPV increases whole-organ oxygen uptake by improving V/Q matching.

## Supporting information

**S1 Text. Appendix.**
(PDF)

**S1 Table. Pulmonary vascular network and perfusion zone morphometry.** Descriptive statistics for each vascular network. Quantities are presented as mean ± standard deviation. The top block summaries the size of the perfusion zones, and the bottom block summarizes the diameters and lengths for each diameter-defined Strahler order.
(PDF)

**S1 Fig. Schematic of our airway model.** Blue boxes denote compartment in our airway model. Airway 1 interacts with the atmospheric air and airway 2. Airway 2 functions as a mixing chamber and interacts with all of the alveolar compartments. Each alveolar compartment exchanges oxygen with a capillary compartment.
(TIF)

**S2 Fig. Effect of tidal volume on oxygenation.** (A) Oxygen tension in the primary vein and mean and standard deviation in the alveolar compartments as a function of tidal volume; (B) Total oxygen flux as a function of tidal volume.
(TIF)

**S3 Fig. Effect of HPV and uniform vasoconstriction on Network A regional oxygen tension distribution.** Leftmost column of networks are without regulation from HPV, middle column of networks are with regulation from HPV, and the rightmost column of networks are with uniform vasoconstriction, where $T^* = 0.8$ is used for the global vascular tone. The black scale bar is 1000 $\mu$m. (A) End-capillary oxygen tension; (B) Alveolar Oxygen tension; (C) Tone from HPV.
(TIF)

**S4 Fig. Effect of airway occlusions on the distribution of vascular tone from HPV.** This figure corresponds to simulations shown in Fig 10. Leftmost network is with 0% of the alveoli occluded, middle network is with 17% of the alveoli occluded, and the rightmost network is with 28% of the alveoli occluded. The first airway occlusion is at the most southwestern perfusion zone, and progressive airway occlusions are made following the structure of the vascular network.
(TIF)

## Author Contributions

**Conceptualization:** Andrew D. Marquis, Daniel A. Beard.

**Data curation:** Andrew D. Marquis.

**Formal analysis:** Andrew D. Marquis, Daniel A. Beard.

**Funding acquisition:** Andrew D. Marquis, Daniel A. Beard.

**Investigation:** Andrew D. Marquis.

**Methodology:** Andrew D. Marquis, Filip Jezek, Daniel A. Beard.

**Project administration:** David J. Pinsky, Daniel A. Beard.

**Software:** Andrew D. Marquis.

**Supervision:** David J. Pinsky, Daniel A. Beard.

**Validation:** Andrew D. Marquis.

**Visualization:** Andrew D. Marquis.

**Writing – original draft:** Andrew D. Marquis.

**Writing – review & editing:** Andrew D. Marquis, Filip Jezek, David J. Pinsky, Daniel A. Beard.

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
