## [Decision Letter · Decision Letter 0]

5 May 2020

Dear Prof. Beard,

Thank you very much for submitting your manuscript "Hypoxic pulmonary vasoconstriction as a regulator of alveolar-capillary oxygen flux: a computational model of ventilation-perfusion matching" for consideration at PLOS Computational Biology.

As with all papers reviewed by the journal, your manuscript was reviewed by members of the editorial board and by several independent reviewers. In light of the reviews (below this email), we would like to invite the resubmission of a significantly-revised version that takes into account the reviewers' comments.

*We feel it will be necessary for you to thoroughly address several significant issues raised by the reviewers for this revision to be successful in the next round, and we'd encourage you to provide comprehensive responses to their concerns.  If these issues are sufficiently and carefully addressed, we'd be happy to consider the revised submission for publication after re-review.   *

We cannot make any decision about publication until we have seen the revised manuscript and your response to the reviewers' comments. Your revised manuscript will then be sent to reviewers for further evaluation.

Sincerely,

Alison Marsden

Associate Editor

PLOS Computational Biology

Mark Alber

Deputy Editor

PLOS Computational Biology

Reviewer's Responses to Questions

**Comments to the Authors:**

Reviewer #1: The authors describe the development of a gas exchange model that accounts for HPV and makes predictions regarding blood flow and gas exchange at various scales. The geometric and gas exchange parameters of the 2D model are selected/fitted to correspond to nominal conditions in the rat lung. The microcirculation is not explicitly represented. The model predicts homogenization of blood flow and oxygen transport at smaller scales, leading to overall improvement in oxygen uptake at the level of the whole lung. The authors are to be commended for providing access to their code.

GENERAL COMMENTS

Please consider adding scale bars to figures including lung lobes - how large was a typical gas exchange unit?

To what extent does tidal ventilation affect gas exchange?

What values were used for the conducted response length constant l_k?

What is the range of vessel sizes simulated and the range of hematocrit values in the smallest vessels?

The authors should comment on the relative increase in total vascular resistance with HPV vs UVC, and to what extent this is a consequence of parameter selection (T values)? What technique was used to estimate the values of T selected for Figure 4? Does the level of hypoxia selected correspond to the maximum sensitivity of HPV in the observed capillary/alveolar PO2 values?

How do the observed values of CV compare to estimates in actual rat lung?

SPECIFIC COMMENTS

L153: Rationale for assumed value of hydraulic conductivity k

Eq 19: Plus sign in denominator

L302: Hypoxic air fraction reads 21% rather than 10%

L336: Should read "proportional to its size"

Reviewer #2: The authors present a model for simulating the response of a network of blood vessels to hypoxic pulmonary vasoconstriction (HPV). The model includes a 2D network geometry, an oscillating ‘alveolar pressure’, a perfusion model, a gas exchange model, and a model for HPV. I have a number of concerns relating to presentation, assumptions, and lack of thorough analysis. The manuscript also needs a good going through for grammar and spelling, and accuracy in the equations and nomenclature. e.g. Palv is used variably for alveolar pressure and (as Palv,i) oxygen partial pressure in the alveoli.

The introduction suggests that this will be a fully coupled model that will allow the authors to explore the impact of HPV on the integrated gas exchange system. But that does not come through in the results, and it is difficult to interpret from the methods as they are missing some important information (details below). The ‘integration’ between airways and vessels is minimal and is not fully described. A time-varying boundary condition for the blood flow model was not given, but I assume that the model was indeed time-varying? The only time-variation for the ‘airway’ model was an oscillation of an alveolar pressure. How does the dynamic blood pressure affect the model? How does the dynamic airway pressure affect the model? (the intro suggests that these are important).

The conclusion of homogenization of blood flow is not representative of the intact lung, where HPV increases heterogeneity of tissue perfusion. You have simulated regional PO2 for a uniform V, and allowed a non-uniform Q to adapt to this. The outcome is that Q becomes adapted to V (i.e. more uniform). However, in the intact lung the V distribution is highly non-uniform, so the conclusion that HPV ‘homogenizes’ Q is only relevant to the artificial scenario considered in the current model. Please interpret your results in the context of intact lung physiology.

The conclusion of increased transit time appears to be because the model predicts a reduction in cardiac output in response to increased resistance. I’m not convinced that this is realistic. And strangely this results in an increased PO2.

This is a model with many components and it is difficult to understand exactly what is necessary. You need to provide a justification of the inclusion of each component, and a thorough analysis of the sensitivity of the model to each component and parameter. There is no evaluation of the contribution of the time-varying alveolar pressure, nor the dynamic model of perfusion. The introduction suggests that these are important factors, but no results are provided that show why they are important. Similarly, there are many tunable parameters in the model. It is essential that you provide a thorough sensitivity analysis to determine their contribution to the model behavior.

A much simpler model with very basic rules could be used to reach the same conclusions. That is, a model that has a simple change in dimension of vessels (increased resistance) in proportion to the downstream alveolar PO2 would give the same conclusions as your model under your scenarios. You therefore need to focus on exactly what is novel in your model, and what new knowledge can be derived from it.

Specific comments

Abstract - “to redirect blood flow to areas”: it is more correct to say “to direct blood flow away from areas that have low PO2”. And same elsewhere throughout the manuscript.

Author summary - Most pulmonary embolism does not result in decreased blood oxygen.

You need to be clear that HPV is an active regulatory mechanism. It is not the primary mechanism for V-Q matching (gravity is, which you mention later for large animals).

Introduction – this is overly long and much of it is superfluous to the actual study. e.g. everything about West’s zones.

Lines 52-57 – but is any of this considered in the model?

Line 19 – blood does not ‘pool’ in the lower lung. That implies that it is stationary.

Line 19 – the marginally higher blood volume in the lower lung does not ‘compress’ it. Tissue compression is a consequence of the weight of the entire lung tissue, which includes blood and extravascular fluid.

Line 23 – there has been a large body of experimental work to understand how V-Q matching affects gas exchange in normal and pathological conditions so the construction of the sentence at Line 22-23 is not correct.

Line 29 – experimental studies have suggested that hypocapnic bronchoconstriction could be an important regulatory mechanism. This has not been proven, but there is some evidence.

Line 42 – what do you mean by the ‘alveolar-capillary boundary’? Are you actually referring to the entire alveolar-capillary interface? Or to the smallest arterioles that supply the alveoli?

Line 107 – and later, throughout: ‘Validated’ is not correct.

Line 144 – are the Fahraeus and Fahraeus-Lindqvist effects relevant to the size of your model vessels?

Line 155 – How were the networks manually calibrated, and how much calibration was required?

Equation 2 – presumably Pv is the transmural pressure for the vessel (internal minus external pressure), where the external pressure is taken as Palv? You need to define this.

Equation 3 – What exactly is Palv, how is it defined, and does it change during the simulation? If this is a dynamic model then I assume that it must change. (Edit – yes, but I only see that because an amplitude is given in Table 1). How does the fluctuation in Palv relate to Equation 12?

Line 182 – how are the capillary segments defined? Do you use a sheet-flow model for the pulmonary microcirculation, or do you assume a simple capillary branch? It seems from Line 212 that you are treating a single ‘capillary vessel segment’ as a whole gas exchange unit (i.e. representing many capillaries with a large surface area). You must provide an explanation and justification in the methods.

Lines 215 and elsewhere – you should refer to oxygen content, not concentration.

Line 256 – As mentioned above, you need to provide explanation of how much manual calibration was required to achieve this.

Line 271 – I appreciate the effort put in to comparing or calibrating to experimental data. But it is not appropriate to call this a “Model validation”. You have introduced a global scaling parameter (T*) to allow reasonable match for one set of studies, and then you optimize other model parameters to provide a good fit to other data. So this is a calibration to provide a reasonable baseline model behavior, rather than a ‘validation’. As such, some of this should be in your methods and not results. Validation requires that you compare against independent data (not used for fitting/calibration), and ideally this should be testing an emergent behavior.

Line 291 – Here you introduce T*. Your pressure-flow relationships are presumably sensitive to the resistance of the capillary bed, but you have not given any information about its dimensions or behavior. The different experiments could also cause different patterns or amounts of capillary bed recruitment. However, by using T* you have assumed that any difference between model and data is purely related to tone, but other factors could be at least as important. Please explain and justify.

I can’t tell whether the simulations were run for normoxia or hypoxia. HPV has been shown to be not important under normoxic conditions, but Equation 22 suggests that you include the effect across all partial pressures. Please explain and justify.

Line 357 – if the mean transit time increases, this implies that the cardiac output has decreased or the blood volume of the model has increased. I assume that you compared cases where the former was equal, so did you have sufficiently different blood volume in the different scenarios to give these different mean transit times? Edit – I have now re-read the manuscript and can see in Table 4 that your total flow decreases in HPV because you use pressure boundary conditions. So of course your transit time will be longer.

Important – can you please explain how your cardiac output can be reduced to approximately one third yet the total oxygen flux increases? (network A and B without and with HPV).

Line 358 – this is only correct if the increased transit time occurs within the gas exchange (your capillary) region. But I believe that in your model the increased transit time is in the larger vessels. Unless I am missing something about the behavior of the capillary in your model (which is not described in the methods).

Line 365 – how can there be more high air flow regions when you have set the air flow to be the same everywhere?

It looks as though you have used P50 for dog. I assume that this is more similar to human than rat, but it is worth commenting on this in the discussion.

Line 465- I strongly disagree that the model has been ‘validated’. It has been calibrated to existing data. And it does not match some important data (e.g. RBC transit time).

Reviewer #3: Authors propose a novel computational model of ventilation-perfusion matching in pulmonary circulation. Work couples three different models- one related to vascular mechanics, one related to oxygen transport and one related to vasoconstriction. It is able to capture data from literature, quantitatively in some and qualitatively in some other. The coupled model is able to show that the hypoxic vasoconstriction homogenizes regional blood flow and alv-capillary oxygen flux.

Strength of the paper is it has elegant ideas related to coupling models and has captured some results in literature successfully. The weakness is the methods section lacks details to understand the model completely.The paper will be stronger with some more rigor on parameter estimation. Comments below:

-- Systematic sensitivity analysis of generated distributed networks to user inputs is missing.

-- There is a scale bar on the distributed networks (Figure 2 and 3). If the networks are non dimensionalized, utility and findings can be extended across species. Can the authors attempt/ give their thoughts on non-dimensionalizing the model?

-- Please include more details on 'manual calibration of network anatomy'. What was used and changed for calibration? How hard was 'manual calibration'? Is it repeatable across users?

-- Airway-vascular oxygen transport section has no references - was the model formulated in this paper? Or is it relying on prior work? This section is hard to to conceptualize and understand with the description provided. Can the authors include more schematics to complement transport equations and descriptions? I am unable to tell airway 1 and airway 2 in Figure 1 (also why "" on "airway 1"?)

-- The formulation of airway-vascular oxygen transport is confusing - the governing equations are in partial pressures, mass of O2 in (moles) (M) and concentration of oxygen. Also, C_o2 is described to be mass of oxygen(mol) in paragraph but is described to be concentration of oxygen prior to equation 20. Can the authors reformulate the equations to keep descriptions consistent through the model?

--Motivation for functional forms used in 'Empirical Hypoxic Pulmonary vasoconstriction' is missing.

--Why is the signal generated by oxygen content exponential - is there biological data to motivate this form?

--Is there evidence of spatial decay of transduced signal in vivo?

--What does the time constant \\tau physiologically represent in equation 24?

--Equation 25 and 26 are elegant ways to couple lumped parameters and vascular tone. Add details so a naive reader understands why you intuited this functional form. Also, is value of T_j bounded for in vivo conditions? -Elaborate

-- Overall, the novel aspects of the 'Methods' are not clear. Three different models are presented -- one related to vascular mechanics, one related to oxygen transport and one related to vasoconstriction -- which sections of these models are novel? Or is it the coupling thats novel? Clarify in methods and add a paragraph in discussion.

-- The coupling with vasoconstriction and vascular mechanics is clear but the coupling between airway and vascular oxygen transport is unclear -- can you add a separate section/figure talking about coupling and what parameters/quantities are exchanged between these models?

--Is the assumption of constant hydraulic conductivity/length through the vascular tree valid in the pulmonary tree? Especially given the heterogeneity in composition and function through the tree. Also, the reference for the value is missing in text.

--In figure 4 - the fit for Network B is not as good as Network A and C. The type of network is also affecting the differences between 'w/o HPV, w/HPV and UVC' across Figure 6,7 and 8. Does this reflect model's ability to capture only a certain kind of network architecture well?

--Figure 5 and Table 2 -- the fits for hypoxic conditions are off by 15-20%, does this mean the model struggles to capture hypoxic conditions? Why? Is there a way to fix it?

Also, acknowledge these shortcomings in Discussion.

--The model has a lot of parameters and very few measurements, can the authors comment on the identifiability of the system? Also, in Table 3 if Tmax and Tmin are correlated does it make sense to estimate only one of them?

-- Given the comment in discussion - have the authors looked at RBC transit time as objective function in their estimation? Its physiologically motivated so makes a great candidate for objective function.

Minor:

--Problem with formatting of Table 1- resize it to avoid overlap with page header and footer

-- thickness(h) in Table 1 is in mmHg?

--The resistance in lumped parameter network is for viscous dissipation - manuscript has used phrases like 'viscous damping', 'viscoelastic damping' etc. -please change it and use these terms carefully.

--line 337 - as 'shown' in Figs 6A, 7A, 8A ....

--line 341 -- check figure numbers for '6D, 7D and 7D'?

**Have all data underlying the figures and results presented in the manuscript been provided?**

Reviewer #1: Yes

Reviewer #2: Yes

Reviewer #3: Yes

PLOS authors have the option to publish the peer review history of their article (what does this mean?). If published, this will include your full peer review and any attached files.

Reviewer #1: No

Reviewer #2: No

Reviewer #3: No
---

## [Decision Letter · Decision Letter 1]

9 Nov 2020

Dear Beard,

Thank you very much for submitting your manuscript "Hypoxic pulmonary vasoconstriction as a regulator of alveolar-capillary oxygen flux: a computational model of ventilation-perfusion matching" for consideration at PLOS Computational Biology.

As with all papers reviewed by the journal, your manuscript was reviewed by members of the editorial board and by several independent reviewers. In light of the reviews (below this email), we would like to invite the resubmission of a significantly-revised version that takes into account the reviewers' comments.

We cannot make any decision about publication until we have seen the revised manuscript and your response to the reviewers' comments. Your revised manuscript is also likely to be sent to reviewers for further evaluation.

Sincerely,

Alison Marsden

Associate Editor

PLOS Computational Biology

Mark Alber

Deputy Editor

PLOS Computational Biology

Reviewer's Responses to Questions

**Comments to the Authors:**

Reviewer #1: The authors are to be commended for the many changes and improvements (including additional simulations) made to the manuscript in response to the submitted reviews. However, in the opinion of this reviewer, addressing the following issues could improve the paper further:

MAJOR COMMENTS

1. Abstract needs to clearly state that 2D network models are used to represent the vascular and airway structures of the lung, and that capillaries are not explicitly represented

2. LL527-534: Authors should reconcile the assumption of fixed Hct with the calculation of vessel radii from Eq 1 using viscosities that are a function of Hct

3. Discussion: Please include a section on the limitations of using fixed pressure boundary conditions for calculations of flow, resistance and oxygen transport as presented in Table 5, including a discussion of how decreased overall flow can improve oxygen flux

MINOR COMMENTS

- Please clarify (in proof) that Fig A1 refers to Figure 11 (and Fig A2 to Fig 12)

- Captions for figures 6, 7, 8, 10 do not indicate dimensions of the scale bar

- L331: Hypoxic air fraction still reads 21%

- L726: For clarity, please explicitly state proportionality constant for tidal volume in Figure A2/12 relative to Q*_V (ml/s)

- LL601-605: Please include in Fig 5 caption that T* = 0.8 was used to simulate UVC for all three networks

- Table 1: What are the nominal values of Q*_P and Q*_V? Is [46] the correct reference for rat values? On what basis (or reference) was a value of 115 mmHg chosen for P_HPV?

- Presumably copyediting will resolve the numerous spelling and typographical errors that remain and/or have been introduced (e.g. LL23, 381, 385, 437, 453, 530, 534, 535, 570, 602, 606, 613, 618, 619, 621, 651, 658, 662, 673, 689, 704, 710)

Reviewer #2: The authors have made a good effort to respond to my questions but have not taken on board my most significant concerns. There are additional issues that need to be addressed in light of the response to some of my questions.

From my original review:

• “Please interpret your results in the context of intact lung physiology”. The primary conclusion (that HPV homogenizes Q) is only relevant to a uniform V, which is not a physiological condition. Either the language needs to change to be clear that this is an artificial test condition, or the authors need to introduce a realistic baseline V (for rat this just needs to have heterogeneity). The authors have run yet more simulations in response to this major concern rather than address it directly. Under normal physiological conditions V is not uniform. So it is entirely misleading to say that HPV ‘homogenizes’ Q. This has not been addressed at all in the responses. In fact, the new results raise a further issue: that under the exact conditions where we expect that HPV will be important the model does not predict a realistic response. You must go through the manuscript thoroughly and very clearly point out that you are considering an artificial situation to test your model. The general non-respiratory reader will not be aware that V is not uniform, so where you say in the abstract “with a uniform ventilation distribution” it is likely to be interpreted as physiological.

• “You need to be clear that HPV is an active regulatory mechanism”. The manuscript still implies that HPV is the primary mechanism regulating V/Q matching. Please fix this. This is throughout the manuscript starting from the abstract (e.g. “model of ventilation-perfusion matching” needs to be modified so the reader understands this is a model of V/Q matching via HPV only).

New concerns arising from responses:

• It is now clear (but only because I asked) that HPV is assumed to be active in normoxia. This goes counter to the current literature. I am willing to be convinced that HPV is more significant in normoxia than experimental studies suggest. But I am concerned that you have started with this opinion and therefore built this assumption into your study without 1. being very upfront and clear that this is the case, and 2. thoroughly testing the assumption. What would happen if you started with a ventilation-perfusion distribution that was heterogeneous but quite well matched (as per the normal lung)?

• How much vascular constriction actually occurs for different PO2 levels? We are not actually shown any results for PO2, only O2 uptake.

• Now that I have clarity on the model being sensitive in normoxia, can you please provide results on the distribution of PO2 (as per the maps in Figures 6,7,8).

• A question that occurs to me in response to the issue on normoxia: if the model adapts to homogenize Q towards the uniform V, presumably the PO2 actually homogenizes. So what sustains the HPV response if PO2 becomes close to ‘normal’?

• When regions of zero V are introduced the model gives a modest change in Q. But this is the condition in which we expect to see HPV being important. You have mentioned this in the limitations, but this is creating a rather backwards story: HPV is the lung’s primary regulatory mechanism to divert flow when there is an abnormality, so we should expect the model to perform well in this situation. Normoxia in a ‘normal’ lung should be the additional scenario.

• Original: “Important – can you please explain how your cardiac output can be reduced to approximately one third yet the total oxygen flux increases? (network A and B without and with HPV).” In response, the authors say that the reduced cardiac output means a longer transit time which in turn means more time for oxygen uptake. But this is only correct for transit times that are less than the time required to reach saturation. This is about 0.2 seconds. For the rest of the transit time the RBCs are not taking up oxygen. Can you please explain how this works in your model. Do you include (or see) saturation? Your model seems to contradict a flow-limited system (where increased flow should increase the net oxygen uptake).

• The authors argue (in their response and in the paper) that previous studies that have suggested HPV is not significant in normoxia are not correct. The argument is not strong and is incomplete. Just because there is persistent heterogeneity of V in hyperoxia does not necessarily mean that there will be regions of persistently low PO2. It is not appropriate to include a criticism of the Arai study on this basis. The conditions that you have simulated (as well as the species) is completely different. So really your study and theirs have no relationship to each other.

Specific points:

The manuscript STILL needs a good going through for grammar and spelling

Abstract: You need to be clear that the model assumes that HPV is active in normoxia and that ventilation is distributed uniformly.

Line 66 –you mean decreased transit time?

Line 81 – I was interested in your citations to support HPV homogenizing Q and RBC transit times. I can’t see that this is claimed in ref 15 (in fact they say “progressive hypoxia leads to an increase in perfusion non-uniformity”). By ‘homogenizing’ are you just meaning that it provides a better match to V?

For the alveolar-capillary O2 flux in Table 5, figures 6,7,8 – what period was this calculated over? This will vary throughout the transit time.

I disagree with the authors’ response to my challenge that a more simple model could have been used. A simple model would indeed give the same conclusions. You could have used an arbitrary branching network with dimensions from the literature and solved a simple Poiseuille model for flow. It need not be an ODE model. By assuming a relationship between alveolar partial pressure and vessel resistance you would have predicted a decrease in flow in areas with low PO2 and an increase to other areas. It would not be very sophisticated but it would have reached the same conclusions!

My original comment “Most pulmonary embolism does not result in decreased blood oxygen.”

My apologies: most acute pulmonary embolism (sub-massive) does not result in a major decrease in blood oxygen because of hyperventilation.

Reviewer #3: Authors have addressed the reviewer comments. Despite comments from another reviewer on the writing during previous round of reviews the manuscript still has a lot of typos and errors. It is distracting from the scientific content. For example:

line 23 - 'in' shouldn't be there

line 185 --- 'do' should be 'to'

equation 21 --- all subscripts are i but its not clear why the denominator has a subscript j!?

line 233 - 'oxygen oxygen'

line 319 --- 'and' upper?

line 331 -- both hypoxic and normoxic air have same % of O2 and CO2?

'airway' is randomly caps and small at multiple places in the manuscript

Resistance is accounting for viscous loss - am not sure why its labeled 'transmural resistance'!

line 469 -- goal 'of' the study

line 597-- 'that that'

check line 613 ('we not have an') and line 618 ('in a true a lung')

The Discussion after revision is now too long.

**Have all data underlying the figures and results presented in the manuscript been provided?**

Reviewer #1: Yes

Reviewer #2: Yes

Reviewer #3: Yes

PLOS authors have the option to publish the peer review history of their article (what does this mean?). If published, this will include your full peer review and any attached files.

Reviewer #1: No

Reviewer #2: No

Reviewer #3: No
---

## [Decision Letter · Decision Letter 2]

4 Mar 2021

Dear Dr Beard,

We are pleased to inform you that your manuscript 'Hypoxic pulmonary vasoconstriction as a regulator of alveolar-capillary oxygen flux: a computational model of ventilation-perfusion matching' has been provisionally accepted for publication in PLOS Computational Biology.

Best regards,

Alison Marsden

Associate Editor

PLOS Computational Biology

Mark Alber

Deputy Editor

PLOS Computational Biology

Reviewer's Responses to Questions

**Comments to the Authors:**

Reviewer #1: 1.As requested, the abstract needs to clearly state that the 2D vascular model extends only to the level of large arterioles and that capillaries are not explicitly represented

2.LL577-582: Authors should comment on the viscosity implications of their constant hematocrit assumption in the range of vessel diameters simulated. The discussion regarding fluid mechanics and pulse wave propagation following this paragraph is not particularly relevant to ventilation-perfusion matching and could be shortened considerably.

3.L378: Authors should provide a reference for the assertion that major increases in pulmonary pressure occur with alveolar oxygen tension values below 80 mmHg

4.Multiple typographical/usage errors remain scattered throughout the text

**Have all data underlying the figures and results presented in the manuscript been provided?**

Reviewer #1: Yes

PLOS authors have the option to publish the peer review history of their article (what does this mean?). If published, this will include your full peer review and any attached files.

Reviewer #1: No

---

## [Editor Report · Acceptance letter]

30 Apr 2021

PCOMPBIOL-D-20-00197R2 

Hypoxic pulmonary vasoconstriction as a regulator of alveolar-capillary oxygen flux: a computational model of ventilation-perfusion matching

Dear Dr Beard,

I am pleased to inform you that your manuscript has been formally accepted for publication in PLOS Computational Biology. Your manuscript is now with our production department and you will be notified of the publication date in due course.

With kind regards,

Andrea Szabo
